# Modeling reservoir surface temperatures for regional and global climate models: a multi-model study on the inflow and level variation effects

Manuel Almeida[1], Yurii Shevchuk[2], Georgiy Kirillin[3], Pedro M. M. Soares[4], Rita M. Cardoso[5], José Matos[6], Ricardo Rebelo[7], António Rodrigues[8], Pedro Coelho[9]

[1, 7, 8, 9] Faculdade de Ciências e Tecnologia, Universidade Nova de Lisboa, Lisboa, 2825 - 516, Portugal
[2] MX Automotive GmbH, Berlin, 13355, Germany
[3] Department of Ecohydrology, Leibniz-Institute of Freshwater Ecology and Inland Fisheries (IGB), Berlin, 12587, Germany
[4, 5] Instituto Dom Luís (IDL), Faculdade de Ciências, Universidade de Lisboa, Lisboa, 1749 - 016, Portugal
[6] Stucky SA, Rue du Lac 33, 1020 Renens, Switzerland

*Correspondence to*: Manuel Almeida (mcvta@fct.unl.pt)

**Abstract.** The complexity of the state-of-the-art climate models requires high computational resources and imposes rather simplified parameterization of inland waters. The effect of lakes and reservoirs on the local and regional climate is commonly parameterized in regional or global climate modeling as a function of surface water temperature estimated by atmosphere-coupled one-dimensional lake models. The latter typically neglect one of the major transport mechanisms specific to artificial reservoirs: heat and mass advection due to in- and outflows. Incorporation of these essentially two-dimensional processes into lake parameterizations requires a trade-off between computational efficiency and physical soundness, which is addressed in this study. We evaluated the performance of the two most used lake parameterization schemes and a machine learning approach on high-resolution historical water temperature records from 24 reservoirs. Simulations were also performed at both variable and constant water level to explore the thermal structure differences between lakes and reservoirs. Our results highlight the need to include anthropogenic inflow and outflow controls in regional and global climate models. Our findings also highlight the efficiency of the machine learning approach, which may overperform process-based physical models both in accuracy and in computational requirements, if applied to reservoirs with long-term observations available. Overall, results suggest that the combined use of process-based physical models and machine-learning models will considerably improve the modeling of air-lake heat and moisture fluxes. A relationship between mean water retention times and the importance of inflows and outflows is established: reservoirs with the retention time shorter than ~100 days, if simulated without in- and outflow effects, tend to exhibit a statistically significant deviation in the computed surface temperatures regardless of their morphological characteristics.

# 1 Introducion

Numerical weather prediction (NWP) and climate modeling are essential tools in research and applied science applications (e.g., Bauer et al., 2015; Forster, 2017; Jacob et al., 2020). Motivated by the need to increase the reliability of climate and weather projections, the core numerical models undergo continuous improvements aiming at the best compromise between model representativity and computational efficiency (Flato et al., 2013). Air-lake heat and moisture fluxes affect the near surface atmospheric layers and are essential to accurate estimation of the future climate or weather forecast. Therefore,

parameterization of inland waterbodies in atmospheric modeling has quickly evolved to increase the accuracy of the land-atmosphere boundary layers (Bennington, 2014; Xue et al., 2017; Wang et al., 2019a).

According to previous studies, the presence of waterbodies affects significantly the turbulent heat exchange with the atmosphere (Philips, 1972; Bates et al., 1993; Niziol et al., 1995; Lofgren, 2006; Notaro et al., 2013; Wright et al., 2013). In northern latitudes, surface waters tend to absorb heat in summer and release it in autumn, damping the temperature fluctuations

in their vicinity and creating both a lag in diurnal and annual cycles of the air temperature, as well as increased precipitation (Dutra et al., 2010; Nordbo et al., 2011; Samuelsson et al., 2010; Subin et al., 2012). Overall, missing the lake and reservoir effects has been shown to deteriorate the simulation results of regional and global climate simulations (Ljungemyr et al., 1996; Long et al., 2007; Deng et al., 2013; Dutra et al., 2010; Samuelsson et al., 2010; Subin et al., 2012; Le Moigne et al., 2016; Irambona et al., 2018).

Waterbodies display larger thermal inertia than the surrounding land areas due to the high specific heat capacity of water and the vertical turbulent heat transport from the water surface to its deeper layers. Furthermore, they absorb a higher fraction of solar radiation than land due to a lower albedo and a higher transparency. The heat storage and thermal characteristics of inland waterbodies, acting primarily but not only through water column stability, are influenced by bathymetry, surface area, turbidity, and ice conditions (Schertzer, 1997; Rouse et al., 2003, Oswald and Rouse, 2004; Magee and Wu, 2017). Surface-heat fluxes,

in particular the evaporation rate, are also affected by advection due to inflows and outflows (e.g., deep-water withdrawal) and by water level (WL) fluctuations (Rimmer et al., 2011; Friedrich et al., 2018). These fluctuations are usually much more pronounced in reservoirs than in natural lakes. Herewith, neglecting of the aforementioned water budget variations may lead to errors in surface heat flux predictions, especially in reservoirs.

The progressive increase of the spatial resolution of general circulation models (GCM) and regional climate models (RCM)

resulted in wide implementation of coupled one-dimensional (1-D) models simulating surface energy fluxes in waterbodies, neglecting however the variation of in-, outflows, and WL. The coupled lake and reservoir models differ among each other mainly by the vertical mixing parameterization, classified into three major categories: eddy diffusion models, turbulence models, and bulk mixed layer models. In eddy diffusion models, vertical turbulent mixing is defined by eddy diffusion, parameterized as a function of velocity and stratification strength in form of the gradient Richardson number (e.g.,

HOSTETLER model, Hostetler and Bartlein, 1990; SEEMOD, Zamboni et al., 1992; LIMNMOD, Karagounis et al., 1993; MINLAKE, Fang and Stefan, 1996; CLM, Oleson et al., 2004; CLM4-LISSS, Subin et al., 2012; WRF-Lake, Gu et al., 2015).

More complex approaches, based on k-ε turbulence model, parameterize eddy diffusion based on the Kolmogorov-Prandtl relationship (Svensson, 1978; Burchard et al., 1999; Goudsmit et al., 2002; Stepanenko and Lykossov, 2005). Bulk mixed layer models rely on the self-similarity concept for the temperature-depth profile in the stratified layer and integral budgets for the mixed and bottom layers (Kraus and Turner 1967, Mironov et al., 2010). The performance of some of these models has already been evaluated in modeling intercomparison studies (e.g., Perroud, 2009; Stepanenko et al., 2010; Stepanenko et al., 2013; Thiery et al., 2016; Huang et al., 2019, Wang et al., 2019b, Guseva et al., 2020, Stepanenko, 2020). Generally, these intercomparison studies evaluated the model performance in application to one to three lakes, usually with very particular morphological characteristics, (e.g., very deep or very shallow), over a limited time period. Overall, the results of these studies had an important impact in the further development of the models. In particular, they highlighted the need for intercomparison research projects that include a larger number of waterbodies and a longer modeling simulation.

Data-driven models such as artificial neural networks (ANN) have not yet been considered for the parameterization of lakes in climate models. Nevertheless, they have been successfully used to estimate mean daily and hourly water temperatures in rivers (e.g., Chenard and Caissie, 2008, Hebert et al., 2014) and in lakes (Sharma et al., 2008, Samadianfard et al., 2016, Read et al., 2019). The approach is particularly advantageous when the modeled processes are complex and nonlinear (Sharma et al., 2008), as in the case of surface water temperatures (SWT). In view of the trade-off between results quality and computational efficiency, data-driven models have potential advantages in estimating the effect of lake inflows/outflows on SWT, that motivates their inclusion into model intercomparison studies.

Currently, the major challenge in the parameterization of lakes and reservoirs in climate models is the need to ensure that the models' response is consistent and accurate considering the wide range of morphological characteristics and the high variability of the meteorological forcing. While incorporation of in- and outflows may crucially improve the quality of model predictions, the increased complexity can restrain extension of process-based models and require alternative data-based approaches.

In this study, we evaluate the importance of the energy transfers due to water inflows and outflows when modeling surface water energy fluxes in artificial reservoirs and elaborate a methodology to improve this essential aspect of RCM and GCM. For this purpose, we (i) model 24 Portuguese reservoirs by using four models: a 2-D model to define a calibrated and validated baseline scenario, two 1-D models without the parameterization of inflows/outflows and an ANN, (ii) assess the modeling error in SWT of lakes (similar to a seepage lake) and reservoirs, potentially associated with atmosphere-lake interactions, and (iii) compare the performance and computational requirements of different approaches to predict the evolution of SWT in lakes (similar to a seepage lake) and reservoirs.

## 2 Study area

Portugal is located in southern Europe and has a typical Mediterranean climate. Maximum daily mean air temperature ranges from 13 ℃ in the central highlands to 25 ℃ in the southeast region. The minimum daily mean air temperature ranges from 5 ℃ in the northern and central regions to 18 ℃ in the south. (Soares et al., 2012a). Complex topography and costal processes

define the spatial and temporal heterogeneity of precipitation, which differs from a relatively wet annual maximum above 2
800 mm/yr in the mountainous northwest to a much drier 400 mm/yr in the tendentially flat southeast (Soares et al., 2012b;
Cardoso et al., 2013).

The 24 reservoirs selected for this study are almost entirely located in mainland Portugal, apart from Alto Lindoso (R19) and
Alqueva (R24) reservoirs, which are shared with neighboring Spain (Fig.1).

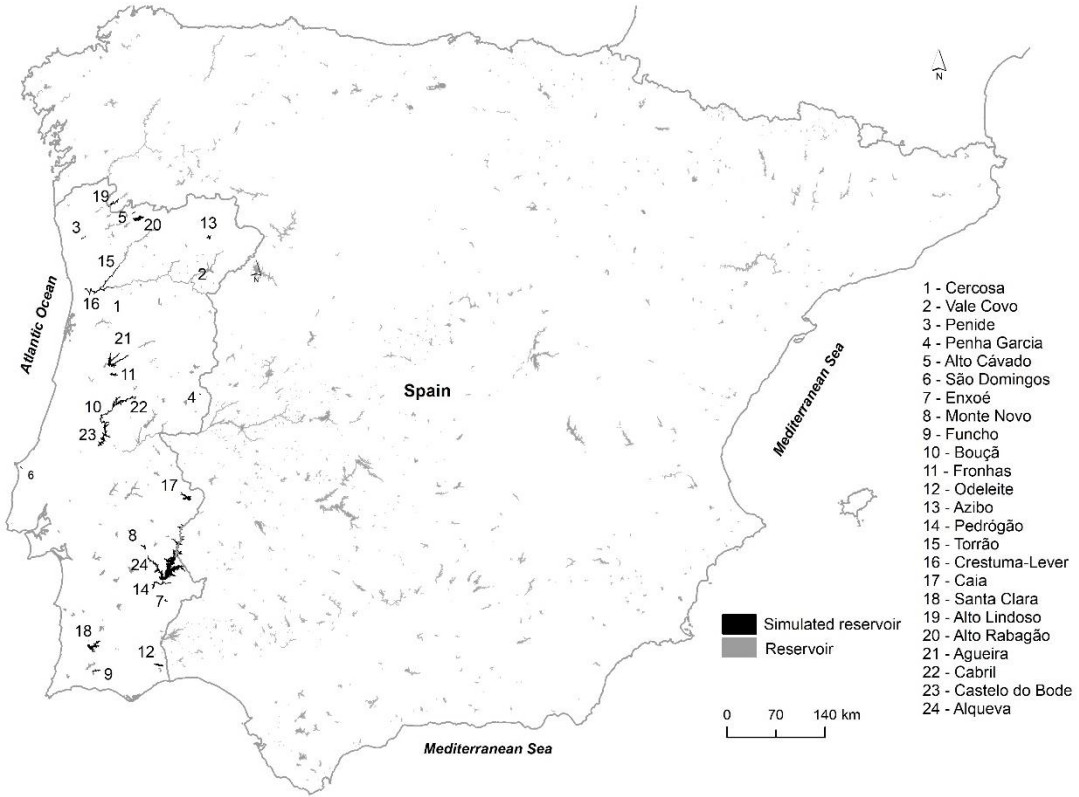

**Figure 1: Location of the simulated waterbodies (ordered according to the simulated mean volume from smallest to largest)**

The reservoirs were selected for the study based on their water residence time (WRT) and morphological characteristics
(volume, depth, surface area) (Table 1). Most of reservoirs are classified as warm monomictic, with a stratified period during
the warmer months (May – September), and one mixing period each year, during the colder part of the year, from October to
April. As exceptions, Cercosa (R1) and Torrão (R15) are weakly stratified, while Penide (R3), Penha Garcia (R4), Enxoé (R7)
and Crestuma-Lever (R16) (a run-of-the-river hydropower scheme, located in the north coastal region) are well mixed during
the entire year.

**Table 1 Morphometric details of the reservoirs and main water uses**

| | Reservoir | Simulation period | Mean volume (hm³) | Full supply volume (FSV) (hm³) | Maximum depth (m) | Mean depth (m)[1] | Water surface area at the FSV (ha) | Watershed area (km²) | Mean inflow (m³.s⁻¹) | Mean water residence time (days)[2] | Main use(s) |
|---|---|---|---|---|---|---|---|---|---|---|---|
| R1 | Cercosa | 1994-2008 | 0.05 | 0.06 | 16.00 | 3.00 | 2.00 | 59.89 | 0.72 | 0.79 | P |
| R2 | Vale Covo | 1994-2008 | 0.10 | 0.20 | 14.00 | 1.67 | 12.00 | 53.41 | 0.00 | 2093.86 | W |
| R3 | Penide | 1989-2008 | 0.11 | 0.50 | 9.00 | 0.72 | 69.00 | 3.73 | 33.63 | 0.04 | P |
| R4 | Penha Garcia | 1989-2008 | 0.38 | 1.10 | 10.00 | 5.39 | 20.40 | 14.73 | 0.05 | 91.50 | W, I |
| R5 | Alto Cávado | 1989-2008 | 1.17 | 3.30 | 21.00 | 6.60 | 50.00 | 101.23 | 4.73 | 2.85 | P |
| R6 | São Domingos | 1994-2008 | 4.61 | 7.90 | 34.00 | 8.23 | 96.00 | 42.04 | 0.07 | 728.27 | W, I |
| R7 | Enxoé | 1998-2008 | 6.48 | 10.40 | 12.00 | 5.07 | 205.00 | 60.54 | 0.27 | 275.67 | W |
| R8 | Monte Novo | 1989-2008 | 10.75 | 15.27 | 19.85 | 5.50 | 277.40 | 260.75 | 0.15 | 830.80 | W, I |
| R9 | Funcho | 1994-2008 | 24.37 | 47.72 | 36.00 | 13.26 | 360.00 | 211.58 | 3.33 | 84.57 | I |
| R10 | Bouça | 1989-2008 | 27.69 | 48.40 | 62.00 | 9.68 | 500.00 | 2601.71 | 44.34 | 7.23 | P |
| R11 | Fronhas | 1989-2008 | 28.08 | 62.10 | 48.00 | 11.61 | 535.00 | 630.46 | 15.05 | 21.59 | P |
| R12 | Odeleite | 1997-2008 | 42.58 | 130.00 | 47.00 | 18.06 | 720.00 | 347.27 | 2.84 | 173.51 | W, I |
| R13 | Azibo | 1989-2008 | 45.62 | 54.50 | 43.00 | 13.29 | 410.00 | 92.56 | 0.79 | 670.93 | W, I |
| R14 | Pedrógão | 2005-2008 | 86.16 | 106.00 | 25.00 | 9.60 | 1104.00 | 59160.00 | 39.94 | 24.97 | P, I |
| R15 | Torrão | 1989-2008 | 91.57 | 124.00 | 56.00 | 19.08 | 650.00 | 3268.28 | 76.44 | 13.86 | P |
| R16 | Crestuma-Lever | 1989-2008 | 101.08 | 110.00 | 13.00 | 8.47 | 1298.00 | 96932.81 | 423.97 | 2.76 | P, W |
| R17 | Caia | 1989-2008 | 112.35 | 203.00 | 44.00 | 10.30 | 1970.00 | 563.26 | 2.45 | 530.77 | W, I |
| R18 | Santa Clara | 1989-2008 | 205.74 | 485.00 | 72.00 | 24.42 | 1986.00 | 519.69 | 2.18 | 1091.19 | P, F, W, I |
| R19 | Alto Lindoso | 1992-2008 | 274.57 | 390.00 | 92.00 | 36.38 | 1072.00 | 1510.93 | 39.44 | 80.58 | P |
| R20 | Alto Rabagão | 1989-2008 | 317.82 | 569.00 | 84.00 | 25.72 | 2212.00 | 106.97 | 9.41 | 390.92 | P |
| R21 | Aguieira | 1989-2008 | 335.59 | 423.00 | 76.00 | 21.15 | 2000.00 | 3063.29 | 87.45 | 44.42 | P, F, W, I |
| R22 | Cabril | 1989-2008 | 344.79 | 719.00 | 120.00 | 35.54 | 2023.00 | 2416.32 | 38.69 | 103.14 | P |
| R23 | Castelo do Bode | 1989-2008 | 859.46 | 1095.00 | 96.00 | 33.27 | 3291.00 | 3964.09 | 64.94 | 153.18 | P, W, F |
| R24 | Alqueva | 2005-2008 | 2974.66 | 4150.00 | 76.00 | 16.60 | 25000.00 | 55289.00 | 38.25 | 900.02 | I, W, P |

P – Power generation; W – Water supply; I – Irrigation; F – Flood prevention

(1) The mean depth results from the division of the mean volume and the mean water surface area.
(2) The mean water residence time is the ratio between the mean volume of the reservoir and the mean inflow.

# 3 Models and data

## 3.1 Models/scenarios

To evaluate the importance of inflow and outflow in SWT simulations, a 2-D numeric model and two 1-D models were applied.

115     Table 2 shows a full description of the scenarios considered in the development of this study.

Since the model validation was limited by the scarcity of temperature profile measurements and observed time series of SWT,

a major challenge of this study consisted in development of realistic baseline scenarios (forcing data and targets; W2 hydrology scenarios) having the necessary continuity and heterogeneity to evaluate the performance of different models. To overcome this limitation, a well-established 2-D model, CE-QUAL-W2 version 3.6 (Cole and Wells, 2008) was validated with observed data and used to create the baseline scenario, forced with daily and hourly meteorological datasets covering a period of 20 years, from 1989 to 2008 (with the exceptions described in Table 1). The 2-D model, forced with daily meteorology and monthly inflows and outflows, was calibrated by minimizing the Mean Absolute Error (MAE) between simulated water temperature profiles, and measurements spanning the period from 1989 to 2008 made in each reservoir, in all cases near the dam (W2 hydrology-D). After each model run, results were compared with the observed data sets and if needed the calibration parameters were retuned manually. The wind wind-sheltering coefficient (WSC) and the extinction coefficient for water were the only parameters modified at each model run. These parameters varied in the range from 0.1 to 1.0 and from 0.25 to 1.0, respectively. Data on the mean water water-extinction coefficient was available for four reservoirs: Bouçã ($\mu$=0.27; $\sigma$=0.05), Crestuma-Lever ($\mu$=0.67; $\sigma$=0.15) - 0.67, Cabril ($\mu$=0.27; $\sigma$=0.05) and Castelo do Bode ($\mu$=0.26; $\sigma$=0.05), therefore they were not calibrated. All 1-D simulations were performed with a constant water-extinction coefficient value of 0.45, corresponding to the reference value suggested by Cole and Wells (2008). According to the eutrophication criteria defined by the OCDE (OCDE, 1982), this value of water transparency is associated with eutrophic unstable systems and is also close to the mean value of 0.37 obtained from the four reservoirs listed above.

An alternative baseline scenario was produced by forcing the model with hourly meteorology (daily values were used for the first one), enabling evaluation of the sub-daily convection effects on the overall results. Both daily and hourly baseline scenarios were designated "W2 hydrology-D" and "W2 hydrology-H", respectively.

To assess the importance of heat transfer and mixing within the waterbodies, the two "W2 hydrology" scenarios were modified and simulated in a steady-state "constant mass budget" excluding precipitation, inflows or outflows. These steady-state scenarios were designated "W2". Apparently, "W2" simulations maintain a constant water level, corresponding to the Full Supply Level (FSL). SWT time series, obtained with both scenarios, W2 hydrology and W2, were compared using statistic error measures (see Sect. 3.3 for more details), assessing the relationship between the reservoir WRT and the error resulting from the neglect of advection due to inflows and outflows (as mentioned in the introduction, a common feature of contemporary GCMs and RCMs).

The baseline scenarios (W2 hydrology) were defined to address the following questions:

i)   How large is the uncertainty associated with the neglect of inflows and outflows?

ii)  How adequate is the performance of simplified 1-D models compared with the state-of-the-art calibrated 2-D model, including parametrization of inflows/outflows and WL variation? What is the relative contribution to the final model error of the in- and outflow neglect vs. neglect of the wind sheltering in meteorological forcing?

iii) Can we identify conceptual differences in representation of the fundamental physical processes (such as differences in the conceptualization of diurnal variations of SWT) by 1-D and 2-D models through the comparison of outputs from daily versus hourly forcing?

iv) How well can ANN simulate the evolution of a reservoir SWT?

The reliability of the baseline scenarios (W2 hydrology) for representation of the reservoir thermal regime has been demonstrated by the model calibration results and is supported by the outcomes of a large number of successful model applications worldwide (*vide* Cole and Wells, 2008). Using 2-D modeling results as a baseline "benchmark" scenario for validating 1-D models allows the isolation of the errors associated with the quality of meteorological forcing and observed data (e.g., water-temperature data sets) while providing the continuity usually unavailable from observational datasets. Hence, the error obtained when comparing 1-D versus 2-D model results is to be regarded as an analytical variable, encapsulating differences among the different scenarios and not the conventional model error (model output *versus* observed data).

While generally accurate, the use of calibrated 2-D models in the scope of complex GCMs and RCMs is restricted by high computational costs. Therefore, the next step of the analysis aimed at evaluation of more computationally effective 1-D models, typically used to parametrize waterbodies within GCMs and RCMs. The reservoirs were simulated with a 1-D eddy diffusion model based in the approach considered by Hostetler and Bartlein (1990) and a 1-D bulk mixed layer model (FLake), both forced with hourly and daily meteorological data. Meteorological datasets considered in the modeling process included: air temperature (ºC); relative humidity (%); wind velocity (m/s); wind direction (rad); cloud fraction (0 to 10) and shortwave-solar radiation (W/m$^2$). These datasets were considered in all models with the following exceptions: wind direction is not considered for 1-D models forcing; the ANN modeling relays in the air-temperature, relative-humidity and wind-velocity datasets only.

**Table 2 Simulation scenarios**

| Scenario | Model | Hydrology (inflows/outflows computation) | Calibration | Calibration parameters | Model time resolution | Additional Comments |
|---|---|---|---|---|---|---|
| W2 hydrology-D (Baseline scenario) | 2-D CE-QUAL-W2 | Yes | Yes | Wind sheltering coefficient; extinction coefficient | Daily | - |
| W2 hydrology-H (Baseline scenario) | 2-D CE-QUAL-W2 | Yes | No | - | Hourly | Equal to W2-hydrology-D, except for the meteorological forcing file |
| W2-D | 2-D CE-QUAL-W2 | No | No | - | Daily | Equal to W2-hydrology-D, but without inflows/outflows (similar to a seepage lake) |
| W2-H | 2-D CE-QUAL-W2 | No | No | - | Hourly | Equal to W2-hydrology-H, but without inflows/outflows (similar to a seepage lake) |
| HLM-D | 1-D Hostetler | No | No | - | Daily | - |
| HLM-H | 1-D Hostetler | No | No | - | Hourly | Equal to HLM-D, except for the meteorological forcing file |
| FLake-D | 1-D FLake | No | No | - | Daily | - |
| FLake-H | 1-D FLake | No | No | - | Hourly | Equal to FLake-D, except for the meteorological forcing file |
| ANN-D | ANN | No | No | - | Daily | - |
| ANN-H | ANN | No | No | | Hourly | Equal to ANN-D, except for the meteorological forcing file |

The eddy diffusion model considers the vertical variation of both eddy diffusion and cross-sectional area. Simulations were undertaken using the maximum depth. In turn, FLake operates with volume-integrated equations. Accordingly, its simulations were performed based on the mean reservoir depth. Results obtained with the 1-D models, without any reservoir-specific calibration, were compared with the baseline scenarios obtained with the 2-D model (W2 hydrology). In addition to the 1-D models, SWT in all the reservoirs was modeled with an artificial neural network (ANN) trained using the momentum gradient-based optimization algorithm (Qian, 1999). SWT from both daily and hourly 2-D baseline scenarios ("W2 hydrology"), covering the period from 1989 to 2004 and the predictor variables described in Table 3, were used to improve the input data dimension (Fig. 2).

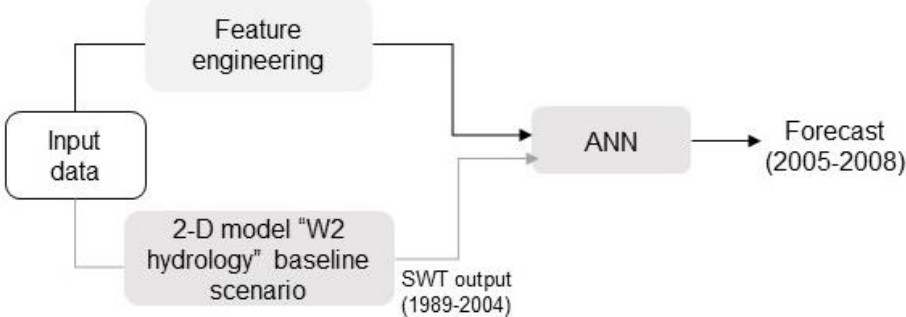

**Figure 2. Schematic and simplified representation of the ANN preparation concept.**

Two different temporal resolutions of the input meteorological data, daily and hourly, were used to train and validate the ANNs. 80% of the data was used for finding optimal network weights (of which 70% were directly applied in the training and 30% were employed in validation). These 80% covered 16 years from 1989 to 2004. The daily discretization resulted in a dataset with N = 5 843 entries, while the hourly discretization produced N = 140 232. The remaining 20% of data had no intervention in the search for optimal network weights and covered the period from 2005 to 2008. This period, considered for the ANN forecast included three dry years, 2005, 2007, 2008 and one wet year, 2006. All the years were warm except for the cold year of 2008. The daily discretization resulted in a dataset with N = 1 461 entries, while the hourly discretization produced N = 35 064. When the reservoir total simulation period (see Table 1) was shorter than 20 years, the dimension of the test dataset was preserved, and the training and validation data sets were reduced. The raw input data used to train the networks included: the SWT obtained from the baseline scenarios prepared with the 2-D model, air temperature (Tair), inflow water temperature (Tbr), dew point temperature (Tdew), relative humidity (HR), and wind velocity (u2). In order to improve model performance additional time series were included as input. They were defined to provide implicit information about seasonal changes (Table 3).

**Table 3 Predictor variables considered for the training and validation of the ANN**

| Temporal sampling | Predictor variables | Total number of predictor variables |
|---|---|---|
| Daily | Meteorological variables = Tair; Tbr; Tdew; HR; $u_2$; <br> Day of year (1 to 365 or 366); week index number (1 to 52 or 53); month index number (1 to 12); <br> Cosine $(2*\pi*($day index number$/365))$; Sine $(2*\pi*($day index number$/365))^*$; <br> Cosine $(2*\pi*($week index number$/52))$; Sine $(2*\pi*($month index number$/52))$; <br> Cosine $(2*\pi*($week index number$/26))$; Sine $(2*\pi*($month index number$/26))$; <br> Cosine $(2*\pi*($month index number$/12))$; Sine $(2*\pi*($month index number$/12))$; <br> Moving average of the meteorological variables with a window of 31 days; <br> Moving variance of the initial meteorological variables with a window of 31 days; | 26 |
| Hourly | Meteorological variables = Tair; Tbr; Tdew; HR; $u_2$ <br> Hour index number, day index number; week index number; month index number <br> Cosine $(2*\pi*($hour index number$/24))$; Sine $(2*\pi*($hour index number$/24))$ <br> Cosine $(2*\pi*($day index number$/365))$; Sine $(2*\pi*($day index number$/365))$ <br> Cosine $(2*\pi*($week index number$/52))$; Sine $(2*\pi*($month index number$/52))$ <br> Cosine $(2*\pi*($week index number$/26))$; Sine $(2*\pi*($month index number$/26))$ <br> Cosine $(2*\pi*($month index number$/12))$; Sine $(2*\pi*($month index number$/12))$ <br> Moving average of the initial meteorological variables with a window of 31 days <br> Moving variance of the initial meteorological variables with a window of 31 days <br> Moving average of the initial meteorological variables with a window of 744 days <br> Moving variance of the initial meteorological variables with a window of 744 days | 39 |

* Cosine and sine series reproduce cyclical yearly variations that can be recognized by the ANN in the inputs without any breaks (as is the case if linear series such as the day of year are used: large jump from 365 to 1 in the beginning of a new year).

The input time series were subsequently standardized through removal of the mean and scaling to unit variance. After the initial tests, which included different network architectures, backpropagation algorithms, regularization strategies, learning rate rules, activation functions, parameter initialization, and the extraction and transformation of features from the input meteorological data, the algorithm was selected, whose results were in the best agreement with SWT from the baseline scenario simulated with the 2-D model (see Sect. 3.3 for more details).

Beside accuracy, the computation time can also be a critical factor in the suitability of models to be used within GCMs or RCMs. The simplified 1-D models considered in this study have a clear advantage regarding the computation time when compared with more complex 1-D and 2-D approaches – a condition that was at the core of their development and that is directly linked with the neglect of inflows and outflows. Recognizing the importance of computational efficiency, the analysis included the quantification of the overall computation times for the process-based physical models and for the ANN. This evaluation was produced with a 2.21 GHz Quad-Core Intel Core i7 (memory: 16 GB 1600 MHz DDR3), by repeating 20 times each simulation.

### 3.1.1 2-D Water quality and hydrodynamic modeling – CE-QUAL-W2

Due to the lateral and layer averaging of the governing equations, the 2-D hydrodynamic and water-quality model CE-QUAL-W2, version 3.6 (Cole and Wells, 2008) is particularly suitable for modeling relatively long and narrow waterbodies, where transverse variations in velocities, temperatures and constituents are negligible. Outlet geometry, outflows and in-pool densities are the input to the selective withdrawal algorithm that calculates vertical withdrawal zone limits. Among the two model options of the withdrawal—line sinks, which are wide in relation to dam width (> 1/10) and point sinks, which are

narrow in relation to dam width (< 1/10)— only point sinks were considered. The point-sink approximation assumes the flow is radial, both longitudinally and vertically (Cole and Wells, 2008). Therefore, for the outflow structure definition, the centerline elevation of the structure was included in the model (Table 4). Additionally, as suggested by Cole and Wells (2008), the algorithm was allowed to retrieve water from the top elevation of the computational grid. The model has been widely applied to stratified water surface systems such as lakes and reservoirs around the world, including Portugal (e.g., Diogo et al.,

2008, Almeida et al., 2015). In order to illustrate the performance of CE-QUAL W2 in reservoir thermal simulations, Cole and Wells (2008), describe the calibration results obtained for 70 reservoirs. In their study, the MAE obtained for all reservoirs was smaller than 1.0 ºC, and for many of them much smaller. The result can be considered outstanding, especially considering that errors were partially related to the quality of the boundary conditions and forcing meteorological data. The Ultimate algorithm was considered as the solution for the numerical transport for temperature and constituents (Cole and Wells, 2008).

Surface heat exchange was computed with the term-by-term algorithm described by Cole and Wells (2008). The reservoirs' bathymetry was defined from 1:25000 topographic charts of the watersheds. Hence, each reservoir computational grid is described by a specific number of branches, segments, and layers (Table 4).




**Table 4. Grid dimensions – 2-D CE-QUAL-W2**

| | Reservoir | Number of branches | Number of segments | Mean segment length, m | Number of layers | Layer height, m | Main outflow centerline elevation, m |
|---|---|---|---|---|---|---|---|
| R1 | Cercosa | 1 | 12 | 100.0 | 11 | 2.0 | 15.0 |
| R2 | Vale Covo | 1 | 7 | 104.5 | 9 | 2.0 | 346.4 |
| R3 | Penide | 1 | 22 | 574.0 | 11 | 1.0 | 15.0 |
| R4 | Penha Garcia | 1 | 10 | 189.7 | 10 | 2.0 | 510.0 |
| R5 | Alto Cávado | 1 | 12 | 519.0 | 28 | 1.0 | 884.2 |
| R6 | São Domingos | 1 | 22 | 204.1 | 19 | 2.0 | 32.2 |
| R7 | Enxoé | 1 | 9 | 500.0 | 8 | 2.0 | 170.0 |
| R8 | Monte Novo | 2 | 18 | 970.6 | 12 | 2.0 | 188.0 |
| R9 | Funcho | 2 | 22 | 907.2 | 21 | 2.0 | 86.0 |
| R10 | Bouçã | 1 | 17 | 1000.0 | 33 | 2.0 | 136.0 |
| R11 | Fronhas | 1 | 22 | 1118.8 | 29 | 2.0 | 100.0 |
| R12 | Odeleite | 4 | 59 | 500.0 | 50 | 1.0 | 10.0 |
| R13 | Azibo | 3 | 21 | 756.4 | 33 | 1.0 | 578.0 |
| R14 | Pedrógão | 2 | 21 | 1828.6 | 13 | 2.0 | 80.0 |
| R15 | Torrão | 1 | 34 | 1000.0 | 36 | 2.0 | 25.0 |
| R16 | Crestuma-Lever | 2 | 98 | 500.0 | 24 | 1.0 | 2.5 |
| R17 | Caia | 2 | 28 | 1000.0 | 25 | 2.0 | 212.0 |
| R18 | Santa Clara | 4 | 57 | 1006.0 | 39 | 2.0 | 120.0 |
| R19 | Alto Lindoso | 2 | 54 | 500.0 | 49 | 2.0 | 250.0 |
| R20 | Alto Rabagão | 2 | 38 | 463.0 | 41 | 2.0 | 800.0 |
| R21 | Aguieira | 3 | 83 | 850.6 | 46 | 2.0 | 83.5 |
| R22 | Cabril | 2 | 76 | 1000.0 | 61 | 2.0 | 220 |
| R23 | Castelo do Bode | 10 | 148 | 735.0 | 48 | 2.0 | 42.0 |
| R24 | Alqueva | 3 | 87 | 2210.8 | 75 | 1.0 | 105.8 |

### 3.1.2 Eddy diffusion model – Hostetler model (HLM)

The governing equation for the 1-D eddy-diffusion model is based on Hostetler and Bartlein (1990):

$$\frac{\partial T}{\partial t} = \frac{1}{A(z)} \frac{\partial}{\partial z} \left\{ A(z)[k_m + K(z,t) \frac{\partial T}{\partial z}] \right\} + \frac{1}{A(z)} \frac{1}{C_w} \frac{\partial[\Phi A(z)]}{\partial z} \tag{1}$$

where T, t, z, A, $k_m$, K, $C_w$, and Φ are: water temperature (⁰C), time (s), depth (m), area (m²), molecular diffusion (1.39·10⁻⁷ m²·s⁻¹), eddy diffusion (m²·s⁻¹), the volumetric heat capacity of water (J·m⁻³·⁰C⁻¹) and a heat source term (W·m⁻²), respectively.

Within the model, eddy diffusion is computed at each depth with the analytical representation developed by Henderson-Sellers (1985) as a function of the 2 m wind velocity ($u_2$), and a latitude-dependent parameter of the Ekman profile.

The surface boundary condition is described by the following equation:

$$\rho C_w [k_m + K(z,t)] \frac{\partial T}{\partial z}\Big|_{z=o} = q_n \tag{2}$$

The net surface heat flux ($q_n$) (W·m⁻²), which is the algebraic sum of solar radiation, atmospheric radiation, latent and sensible heat fluxes and back radiation, was computed with the equilibrium temperature approach defined by Edinger et al., (1968), while latent and sensible heat fluxes were computed explicitly from surface water temperature with the same expressions defined in Cole and Wells (2008). In this study the heat transferred from the sediments to the water column has been neglected. Accordingly, the bottom boundary condition takes the following form:

$$\rho C_w [k_m + K(z,t)] \frac{\partial T}{\partial z}\Big|_{z=maxdepth} = 0 \tag{3}$$

The solution of the heat diffusion equation was obtained resorting to the implicit numeric Crank-Nicholson scheme with centered differences in space and time. Convective mixing is conceptualized by a full-depth mixing scheme that detects buoyancy-induced instabilities and mixes all layers from the surface down to the unstable layer while preserving the available energy. HLM has predicted accurately water temperature profiles of several lakes located in the United States (e.g., Hostetler and Bartlein, 1990; Hostetler and Giorgi, 1995) and a modified version of the model is currently used in the Community land model that is coupled with the International Centre for Theoretical Physics (ICTP) Regional Climate Model, version 4 (RegCM4) (Bennington et al., 2014). The model governing equation and the parameterization of eddy diffusion is also the base of the 1-D lake model included in the Weather Research and Forecasting (WRF) model (LISSS) (Xiao et al., 2016).

### 3.1.3 FLake model

The FLake model was developed for use in NWP and is currently implemented in several NWP models, for example: the Consortium for Small-scale Modeling (COSMO) from the German Weather service (Mironov et al., 2010); the High Resolution Limited Area Model (HIRLAM), from the Finnish Meteorological Institute; the Icosahedral Nonhydrostatic (ICON), from the German Weather service; or the Integrated Forecast System (IFS), from the European Centre for Medium-Range Weather Forecasts. The model has also been used to evaluate the effects of lakes in the climate system (Gula and Peltier, 2012; Le Moigne et al., 2016) and in the future scenarios for lake water temperature and mixing regimes (Kirillin, 2010; Shatwell et al., 2019). Conceptually, the FLake belongs to the family of "bulk" mixed layer models (Kraus and Turner 1967), widely used in lake studies (e.g. DYRESM: Magee and Wu, 2017; GLM: Hipsey et al., 2019; CSLM: MacKay, 2019). A distinguishing feature of the FLake consists in the extension of the "bulk" approach on the stratified part of the lake water

column from the base of the mixed layer down to the lake bottom. The extension relies on the concept of the thermocline self-similarity (Kitaigorodskii and Miropolsky, 1970), i.e. preserved shape of the temperature profile in the stratified part of the water column. In FLake, a waterbody can be represented as a two-layered system, where the vertical profile of water temperature is parameterized as:

$$T = \begin{cases} T_s & at & 0 \leq z \leq h \\ T_s - (T_s - T_b) \cdot \Phi_T(\zeta) & at & h \leq z \leq D \end{cases} \tag{4}$$

where $z$ is the vertical coordinate, $h$ is the surface mixed layer depth, $D$ is the lake depth, $T_s$ is the mixed layer temperature and, $T_b$ the temperature at the water-sediment interface in the bottom and $\Phi_T(\zeta)$ is the self-similarity function (dimensionless temperature).

### 3.1.4 Artificial Neural-Network

The prototyping and building of the ANN was implemented with the python library NeuPy (Shevchuk, 2015). NeuPy uses Tensorflow (an open-source platform for machine learning) as a computational backend for deep learning models (Abadi et al., 2015). The momentum algorithm used in the selected ANN is an iterative first order optimization method that uses gradient calculated from the average loss of a neural network (usually the mean squared error). The "momentum" applies to information about past gradients during the training in the way that promotes a gradual transition in the balance between stability and rate of change (Qian, 1999).

In addition to the input and output layers, the chosen network has one hidden layer with 24 nodes. Each of these used Rectified Linear Activation Functions (ReLu). Training data was shuffled before training, weights were randomly initiated, and the loss function included the MSE (see further below) to measure the accuracy of the results. Additionally, it used L2 regularization (the adopted regularization constant was 0.002). The step decay algorithm was used to regularize the learning rate (initial value = 0.05, reduction frequency = 750).

### 3.2 Forcing and calibration data

The W2 hydrology scenario was forced by monthly records of inflow and discharge for the period 1989-2008. To characterize inflow daily temperatures of 70 reservoir tributaries, a total of 31 air and water temperature linear regressions were additionally computed from 8 492 pairs of values ($\bar{x} = 274$; SD $\pm$ 565). The mean $R^2$ considering all regressions varied from 0.75 to 0.90 ($\bar{x} = 0.82$; SD $\pm$ 0.03). The calibration of the baseline scenario was performed on 677 water temperature profiles ($\bar{x} = 53$ per reservoir) and 3 738 surface observed values ($\bar{x} = 163$ per reservoir). The hydrometric and water quality data was collected by the Portuguese Environmental Agency, Energies of Portugal, and the Alqueva Development and Infrastructure Company and is available from: www.snirh.ambiente.pt.

A deeper insight into the relationship between the air and surface temperatures may be obtained by application of more detailed semi-stochastic models (Toffolon and Piccolroaz 2015), while the effects of the reservoir volume (depth) and the flow would require specific attention in this case (Calamita et al., 2021).


### 3.2.1 Meteorology

The hourly meteorological datasets of air temperature, relative humidity and wind velocity used as forcing of reservoir models were produced by a high-resolution (9 km horizontal grid spacing) simulation with the Weather Research and Forecasting model (WRF; Skamarock et al., 2008), forced by 20 years of ERA-Interim reanalysis (1989-2008), nested in a domain with a

27 km x 27 km cell size. A more detailed description of the model set-up and simulation results are provided by Soares et al., (2012a) and Cardoso et al., (2013). The WRF hindcast simulation results were extensively validated for inland surface variables, namely: temperatures and precipitation in Portugal (Soares et al., 2012a), Iberian precipitation (Cardoso et al., 2013), and wind (Soares et al., 2014; Rijo et al., 2018; Nogueira et al., 2019). Cloud cover datasets were derived from mean monthly values described in the climatological normal of Portugal (1951-1980), while solar shortwave radiation was computed with an

algorithm based on the EPA method (Thackston and Parker, 1971). Cloud cover reduction of shortwave radiation uses the approach defined by Wunderlich (1972). The daily meteorological datasets, also used to force the models, correspond to the daily mean values obtained from the hourly meteorological datasets.

### 3.3 Evaluation metrics

Model assessment was undertaken relying primarily on the mean bias (Bias), the mean absolute error (MAE), the root mean square root error (RMSE), the centered root mean square error (RMSEc), the coefficient of determination ($R^2$) and the Kling-Gupta efficiency (KGE) (Kling et al., 2012). The metrics were computed with the following equations, where $m_i$ and $o_i$ are the modeled and observed values, and $\bar{m}$ and $\bar{o}$ are their means:

$Bias = \bar{m} - \bar{o}$                                                    (5)

$$\text{MAE} = \frac{1}{N}\sum_{i=1}^{N}|m_i - o_i| \tag{6}$$

$$\text{RMSE} = \sqrt{\frac{1}{N}\sum_{i=1}^{N}(m_i - o_i)^2} \tag{7}$$


$$\text{RMSEc} = \sqrt{\frac{1}{N}\sum_{i=1}^{N}\left((m_i - \bar{m}) - (o_i - \bar{o})\right)^2} \tag{8}$$

$$R^2 = \frac{\sum_{i=1}^{N}(m_i - \bar{o})^2}{\sum_{i=1}^{N}(o_i - \bar{o})^2} \times 100 \tag{9}$$

$$KGE = 1 - \sqrt{(r-1)^2 + \left(\frac{\sigma_m}{\sigma_o} - 1\right)^2 + \left(\frac{\mu_m}{\mu_o} - 1\right)^2} \tag{10}$$

where $r$ is the Pearson coefficient, $\sigma_m$ is the standard deviation of the modeled values, $\sigma_o$ is the standard deviation of the observed values, $\mu_m$ is the modeled values mean and $\mu_o$ is the modeled values mean.

When assessing differences between the models, $m_i$ and $o_i$ are the values obtained for reservoir and for lake simulations,
respectively.

## 4 Results

### 4.1 Models calibration/validation

The wind sheltering coefficient reducing the wind effect on the surface fluxes was found to be the most relevant calibration parameter for the 2-D model (W2 hydrology-D scenario). The overall mean value of the wind sheltering coefficient was of
0.6, with a minimum value of 0.1 in Bouçã (R10) and a maximum of 1.0 in Fronhas (R11), Pedrogão (R14), Aguieira (R21) and Alqueva (R24) reservoirs. The light extinction coefficient was also adjusted during calibration with its value varying from 0.25 to 1.0 ($\bar{x} = 0.38$; SD $\pm$ 0.22). Other coefficients, involved in the water temperature calibration, had a significantly smaller effect and were kept with their default values: 1 m$^2 \cdot$s$^{-1}$ for longitudinal eddy viscosity and diffusivity; 70 m$^2 \cdot$s$^{-1}$ for Chézy coefficient, and 0.45 for solar radiation percentage absorbed in the surface layer (β). The water temperature profiles and surface
temperature time series obtained at the downstream edge of the reservoirs (near the dams) suggested that the reservoirs were reasonably well simulated by the 2-D model (W2 hydrology-D scenario) after the calibration forced with daily meteorology. When comparing the model results with a total of 3 608 observed surface temperature values (Fig. 3a), the MAE varied from 0.87 ºC to 3.54 ºC ($\bar{x} = 1.89$ ºC; SD $\pm$ 0.40 ºC) the RMSE varied from 1.49 ºC to 4.58 ºC ($\bar{x} = 2.41$ ºC SD $\pm$ 0.50 ºC) and the KGE values varied from 0.61 to 0.96 ($\bar{x} = 0.78$; SD $\pm$ 0.09). The three highest RMSE values were obtained for reservoirs with
short WRT, suggesting that the major source of inaccuracy was attributed to the inflow temperatures (R11: 4.58 ºC, WRT: 21.6 days; R1: 3.44 ºC, WRT: 0.79 days; R4: 3.44 ºC, 91.50 days). For the 677 observed water temperature profiles (Fig. 3b), the MAE varied from 1.64 ºC to 2.62 ºC ($\bar{x} = 2.14$ ºC; SD $\pm$ 1.35 ºC) (Fig. 3c) the RMSE varied from 1.77 ºC to 3.52 ºC ($\bar{x} = 2.46$ ºC; SD $\pm$ 1.49 ºC) (Fig. 3d) and the KGE values varied from 0.62 to 0.76 ($\bar{x} = 0.71$; SD $\pm$ 0.04) (Fig. 3e). The results show that a KGE value above 0.6 describes a reasonable fit between both datasets.
Additionally, daily and hourly SWT results were compared with the observed SWT values in order to assess the performance of the different models and the influence of the model time resolution. Simulations with the daily time step had a similar

accuracy in all models (Table 5), with the HLM model results being slightly closer to the observed time series. Daily metrics were obtained by comparing SWT values observed at a specific hour in the reservoirs with the daily averages obtained with the model. Therefore, they tend to level the metrics results for each model, in particular the bias (Table 5). In simulations with

the hourly time step, the 2-D model performed expectedly the best among the process-based models, highlighting the robustness of the baseline scenario (W2 hydrology-H). FLake had a worse performance than HLM, considering the hourly results, which can be attributed to differences in the conceptualization of diurnal variations of SWT. Complete mixing within the mixed layer of FLake model reduced the diurnal temperature variations (Martynov et al., 2010). The differences in the diurnal SWT variability were observed across all reservoirs.

The ANN performed best in terms of similarity to observations. The results obtained for each dataset show that the RMSE obtained with the 2-D model and with the ANN had less variations across all reservoirs than the results obtained with the 1-D models (Fig. 4). This result can be attributed to the wind-forcing treatment by 1-D models. The latter do not consider the wind-sheltering effect, which was the most relevant parameter for calibration of the 2-D model, reducing the wind velocity by around 34%. The response to wind stress of elongated reservoirs depends strongly on whether the dominant wind is directed across or

along the reservoir main axis (Mackay, 2019). Therefore, wind direction can significantly affect SWT predictions by influencing surface mass and heat fluxes, which is evaluated in more detail in Sect. 4.3. Additionally, the comparison of W2 hydrology and W2 scenarios results suggests that the SWT of reservoirs R3, R10 and R22 were particularly affected by in- and outflows and/or water level variations. The difference of RMSE values between W2 hydrology and W2 scenarios reached 2.7 °C, 1.2 °C and 0.9 °C, respectively (Fig. 4).

The ensemble analysis of the results obtained with the 1-D models for the period 2005-2008 (Fig. 4d) shows that the models had a similar performance. Overall, results highlight the large interannual variability of reservoir SWT and emphasize the difficulties that arise when modeling these systems.

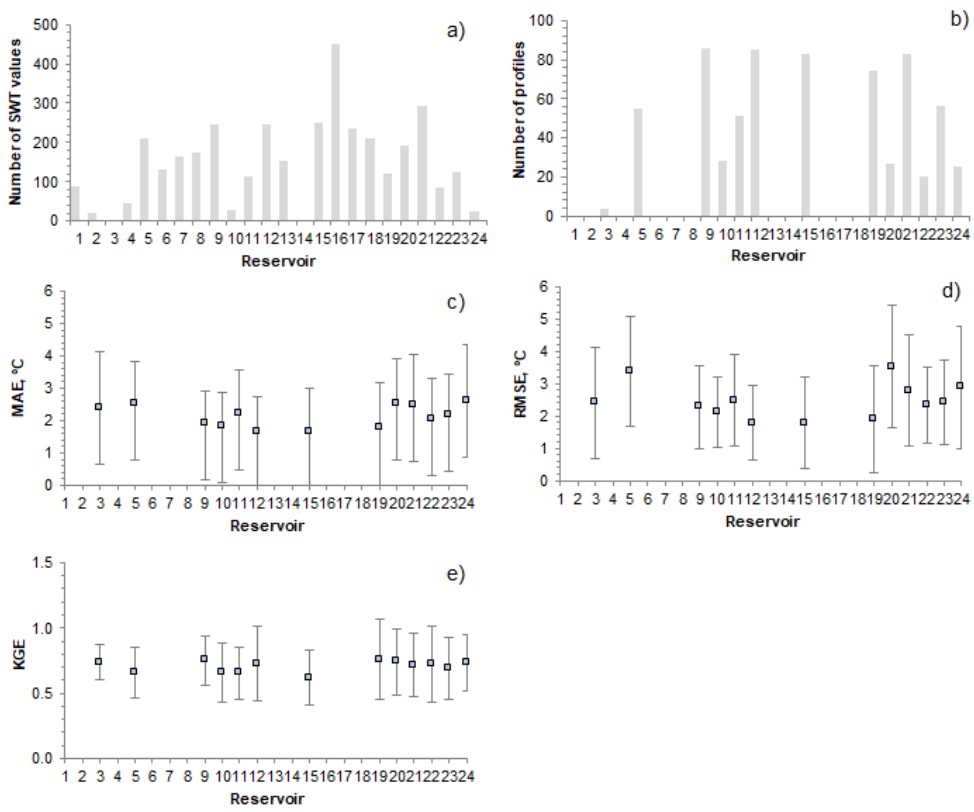

**Figure 3. Number of SWT values (a) and number of water temperature profiles (b) observed in the 24 reservoirs. MAE and standard deviation (c) RMSE and standard deviation (d), KGE and standard deviation (e) between 2-D baseline scenario (W2 hydrology; daily meteorology) simulations and observed water temperature profiles**

**Table 5. MAE, RMSE, bias and KGE (with standard deviation) between observed SWT values and SWT values obtained with all models, forced with daily and hourly meteorology for the 24 waterbodies**

| Time Period | Model | MAE, °C (mean) | | RMSE, °C (mean) | | Bias, °C (mean) | | KGE | |
|---|---|---|---|---|---|---|---|---|---|
| | | Daily | Hourly | Daily | Hourly | Daily | Hourly | Daily | Hourly |
| | W2 hydrology | 1.89 (± 0.40) | 1.85 (± 0.46) | 2.45 (± 0.50) | 2.41 (± 0.49) | 0.20 (± 0.77) | 0.71 (± 0.78) | 0.78 (± 0.10) | 0.81 (± 0.07) |
| | W2 | 2.13 (± 0.69) | 2.16 (± 0.71) | 2.71 (± 0.75) | 2.74 (± 0.75) | 0.32 (± 1.13) | 0.81 (± 1.16) | 0.72 (± 0.23) | 0.75 (± 0.22) |
| **1989-2008** | HLM | 1.72 (± 0.62) | 1.93 (± 0.64) | 2.27 (± 0.62) | 2.46 (± 0.64) | 0.19 (± 1.26) | 0.76 (± 1.04) | 0.85 (± 0.11) | 0.82 (± 0.14) |
| | FLake | 1.75 (± 0.56) | 2.67 (± 0.72) | 2.32 (± 0.56) | 3.20 (± 0.70) | 0.74 (± 0.92) | 2.16 (± 1.07) | 0.84 (± 0.12) | 0.77 (± 0.15) |
| | ANN | - | - | - | - | - | - | - | - |
| | W2 hydrology | 1.89 (± 0.40) | 1.81 (± 0.35) | 2.45 (± 0.50) | 2.33 (± 0.70) | 0.20 (± 0.77) | 0.81 (± 0.80) | 0.79 (± 0.10) | 0.82 (± 0.07) |
| | W2 | 2.08 (± 0.74) | 2.14 (± 0.64) | 2.62 (± 1.02) | 2.72 (± 0.88) | 0.34 (± 1.02) | 0.80 (± 1.22) | 0.74 (± 0.23) | 0.73 (± 0.27) |
| **2005-2008** | HLM | 1.75 (± 0.69) | 1.94 (± 0.66) | 2.25 (± 0.85) | 2.53 (± 0.75) | 0.34 (± 1.29) | 0.88 (± 1.11) | 0.85 (± 0.11) | 0.80 (±0.13 ) |
| | FLake | 1.66 (± 0.53) | 2.63 (± 0.67) | 2.22 (± 0.86) | 3.16 (± 0.77) | 0.74 (± 0.92) | 2.13 (± 1.07) | 0.84 (± 0.12) | 0.78 (± 0.13 ) |
| | ANN | 1.77 (± 0.48) | 1.78 (± 0.44) | 2.28 (± 0.71) | 2.24 (± 0.58) | 0.10 (± 0.93) | 0.60 (± 1.03) | 0.76 (± 0.20) | 0.83 (± 0.07) |

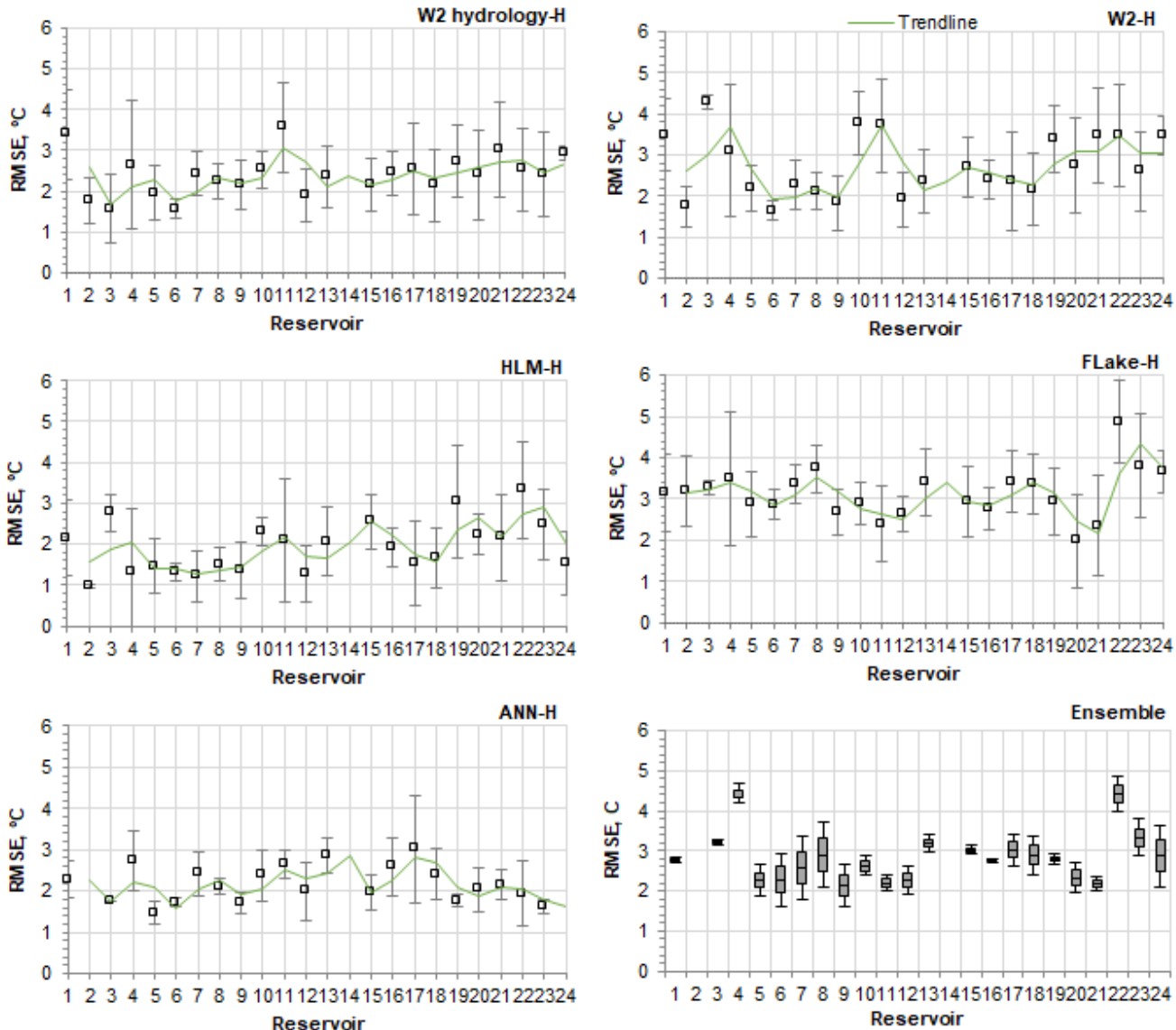

**Figure 4. RMSE between simulated and observed SWT time series considering hourly meteorology for the 24 waterbodies and standard deviation (time period: 1989-2008, except ANN - time period: 2005-2008). The ensemble graphic describes the mean, maximum and minimum RMSE value obtained with the 1-D models – time period: 2005-2008 (box-plot description: maximum, 75th percentile, median, 25th percentile, and minimum).**


## 4.2 Model intercomparison: 1-D models and the ANN

### 4.2.1 Model accuracy

In order to evaluate the consistency and accuracy of the models, the SWT time series were compared with the baseline scenario
W2 hydrology (Table 6). When forced with hourly meteorological data, the ANN reduced significantly the error in SWT predictions for the period 2005-2008 (Fig. 5). This fact emphasizes both the potential of data-driven models to simulate the SWT and the importance of the temporal resolution of the training data sets. Overall, the ANN results remained consistent across both dry and wet seasons, reducing the annual RMSE to 0.86 ºC (± 0.31; daily meteorology) and to 0.71 ºC (± 0.21; hourly meteorology), as well as the interannual variability of RMSE. Accordingly, the KGE values are above 0.96 (Table 6).
Results of both 1-D models were similar to each other (Fig. 5), with both models reproducing well the seasonal variation of SWT and exhibiting a significant variation between the simulations performed with daily and hourly meteorological forcing and during the wet and dry seasons. Nevertheless, FLake and HLM demonstrated a reasonable performance (Table 6), similar to that reported in previous studies (Stepanenko et al., 2010, Stepanenko et al., 2013, Thiery et al., 2016, Guo et al., 2021). The two 1-D models revealed a contradictory behavior with respect to the temporal resolution. In contrast to the HLM, the
FLake model had a slightly better performance with the daily than with the hourly meteorological input, which can also be attributed to differences in the conceptualization of diurnal variations of SWT. Therefore, with daily simulations, these differences between models are much less pronounced.

Considering the bias values obtained for each reservoir (Fig. 5), FLake and the HLM underestimated the SWT in 83% and in 54% of cases, respectively. The negative SWT bias can be primarily ascribed to the overestimation by 1-D models of the wind
stress effect on the surface heat flux due to ignoring the wind direction variability over wind-sheltered elongated reservoirs. The lower bias in the HLM than in FLake is more consistent with the 34% wind velocity reduction obtained in the 2-D model calibration, suggesting the FLake performance was affected by other factors, such as the diurnal SWT variability.

The analysis of the mean annual RMSE obtained with the HLM-H, FLake-H and with the W2-H scenarios considering the hourly meteorology indicate that Penide reservoir (R3), with a WRT of approximately 0.04 days, had the highest mean RMSE,
clearly highlighting the relevance of inflows and outflows in SWTs. HLM had a worse performance for reservoirs R3, R11, R14, R1 and for the six deepest reservoirs, R19, R20, R21, R23, R22 and R24, which indicates that the vertical heat diffusion was not optimally computed (Fig. 5b). Specifically, the explicit approximation of convective mixing in the HLM model by convective adjustment of unstable temperature profiles is apparently too rough, to simulate convective mixing in deep lakes (Bennington et al., 2014). However, it is relevant to mention that the KGE values obtained for 1-D models indicate that, overall,
they performed well (Table 6).

The analysis of the ensemble of RMSE results obtained with all models (Fig. 5e) reveals a high variability among SWT predictions by different models. In general, the performance of 1-D models suggests that their simplified nature and the neglect of inflows/outflows can impose high uncertainties in SWT predictions (Table 6).

**Table 6 Evaluation of model performances. 2-D baseline scenario (W2 hydrology) *versus* simulated SWTs with the exclusion of inflows and outflows (W2), HLM, FLake and ANN. Models forced with daily and hourly meteorology for the 24 waterbodies (RMSE, bias and KGE with the standard deviation)**

**Annual**

| Model | Period | RMSE, ºC (mean) | | RMSE, ºC (max) | | RMSE, ºC (min) | | Bias, ºC (mean) | | KGE | |
|---|---|---|---|---|---|---|---|---|---|---|---|
| | | Daily | Hourly | Daily | Hourly | Daily | Hourly | Daily | Hourly | Daily | Hourly |
| W2 | 1989-2008 | 1.23 (± 1.13) | 1.20 (± 1.03) | 5.03 | 4.54 | 0.17 | 0.19 | -0.04 (± 0.71) | 0.08 (± 0.63) | 0.88 (± 0.24) | 0.88 (± 0.15) |
| | 2005-2008 | 1.24 (± 1.09) | 1.22 (± 1.00) | 5.02 | 4.57 | 0.19 | 0.21 | -0.02 (± 0.70) | 0.10 (± 0.63) | 0.88 (± 0.23) | 0.88 (± 0.15) |
| Hostetler | 1989-2008 | 2.04 (± 0.80) | 1.93 (± 0.72) | 5.07 | 4.39 | 0.83 | 1.12 | -0.21 (± 1.18) | -0.07 (± 1.10) | 0.80 (± 0.15) | 0.82 (± 0.11) |
| | 2005-2008 | 2.08 (± 0.78) | 1.98 (± 0.71) | 4.97 | 4.35 | 0.83 | 1.17 | -0.17 (± 1.24) | -0.03 (± 1.16) | 0.80 (± 0.15) | 0.82 (± 0.11) |
| Flake | 1989-2008 | 1.85 (± 0.58) | 2.31 (± 0.73) | 3.80 | 3.93 | 0.93 | 1.02 | 0.36 (± 0.95) | 1.54 (± 1.03) | 0.84 (± 0.15) | 0.82 (± 0.12) |
| | 2005-2008 | 1.83 (± 0.61) | 2.19 (± 0.81) | 3.78 | 3.91 | 0.92 | 0.83 | 0.32 (± 1.00) | 1.38 (± 1.11) | 0.84 (± 0.15) | 0.82 (± 0.12) |
| ANN | 1989-2008 | - | - | - | - | - | - | - | - | - | - |
| | 2005-2008 | 0.86 (± 0.31) | 0.71 (± 0.21) | 1.61 | 1.25 | 0.52 | 0.24 | -0.06 (± 0.19) | -0.06 (± 0.13) | 0.98 (± 0.03) | 0.96 (± 0.03) |

| | | **Wet season** | | | | **Dry season** | | | |
|---|---|---|---|---|---|---|---|---|---|
| | | RMSE, ºC (mean) | | Bias, ºC (mean) | | RMSE, ºC (mean) | | Bias, ºC (mean) | |
| Model | Period | Daily | Hourly | Daily | Hourly | Daily | Hourly | Daily | Hourly |
| W2 | 1989-2008 | 1.21 (± 0.86) | 1.20 (± 0.86) | 0.19 (± 0.78) | 0.34 (± 0.76) | 1.18 (± 1.40) | 1.14 (± 1.23) | -0.38 (± 1.46) | -0.18 (± 1.32) |
| | 2005-2008 | 1.23 (± 0.82) | 1.22 (± 0.83) | 0.32 (± 0.79) | 0.34 (± 0.84) | 1.18 (± 1.38) | 1.16 (± 1.20) | -0.36 (± 1.43) | -0.15 (± 1.30) |
| Hostetler | 1989-2008 | 1.92 (± 0.67) | 1.77 (± 0.57) | -1.16 (± 1.26) | 0.19 (± 1.11) | 2.05 (± 1.11) | 1.99 (± 1.02) | 0.56 (± 1.77) | -0.34 (± 1.69) |
| | 2005-2008 | 1.97 (± 0.69) | 1.85 (± 0.63) | -1.03 (± 1.28) | 0.25 (± 1.23) | 2.08 (± 1.11) | 2.02 (± 1.00) | 0.68 (± 1.79) | -0.30 (± 1.75) |
| Flake | 1989-2008 | 1.46 (± 0.62) | 2.00 (± 0.84) | -0.23 (± 0.82) | 1.40 (± 0.96) | 2.09 (± 0.81) | 2.43 (± 0.98) | 0.83 (± 1.64) | 1.56 (± 1.66) |
| | 2005-2008 | 1.47 (± 0.60) | 1.95 (± 0.81) | -0.23 (± 1.00) | 1.38 (± 0.94) | 1.98 (± 0.87) | 2.32 (± 1.03) | 0.78 (± 1.63) | 1.38 (± 1.76) |
| ANN | 1989-2008 | - | - | - | - | - | - | - | - |
| | 2005-2008 | 0.81 (± 0.28) | 0.69 (± 0.23) | -0.11 (± 0.20) | -0.11 (± 0.18) | 0.88 (± 0.35) | 0.73 (± 0.23) | 0.02 (± 0.23) | -0.02 (± 0.17) |

| | | **Wet season** | | **Dry season** | |
|---|---|---|---|---|---|
| | | KGE | | KGE | |
| Model | Period | Daily | Hourly | Daily | Hourly |
| W2 | 1989-2008 | 0.83 (± 0.21) | 0.86 (± 0.19) | 0.91 (± 0.12) | 0.91 (± 0.11) |
| | 2005-2008 | 0.84 (± 0.20) | 0.86 (± 0.18) | 0.91 (± 0.12) | 0.90 (± 0.11) |
| Hostetler | 1989-2008 | 0.77 (± 0.18) | 0.81 (± 0.15) | 0.82 (± 0.09) | 0.84 (± 0.08) |
| | 2005-2008 | 0.79 (± 0.16 ) | 0.81 (± 0.15) | 0.84 (± 0.09) | 0.82 (± 0.08) |
| Flake | 1989-2008 | 0.83 (± 0.16) | 0.81 (± 0.16) | 0.86 (± 0.07) | 0.83 (± 0.07) |
| | 2005-2008 | 0.84 (± 0.14) | 0.81 (± 0.15) | 0.86 (± 0.07) | 0.83 (± 0.08) |
| ANN | 1989-2008 | - | - | - | - |
| | 2005-2008 | 0.98 (± 0.03 ) | 0.96 (± 0.03 ) | 0.98 (± 0.03) | 0.96 (± 0.03) |

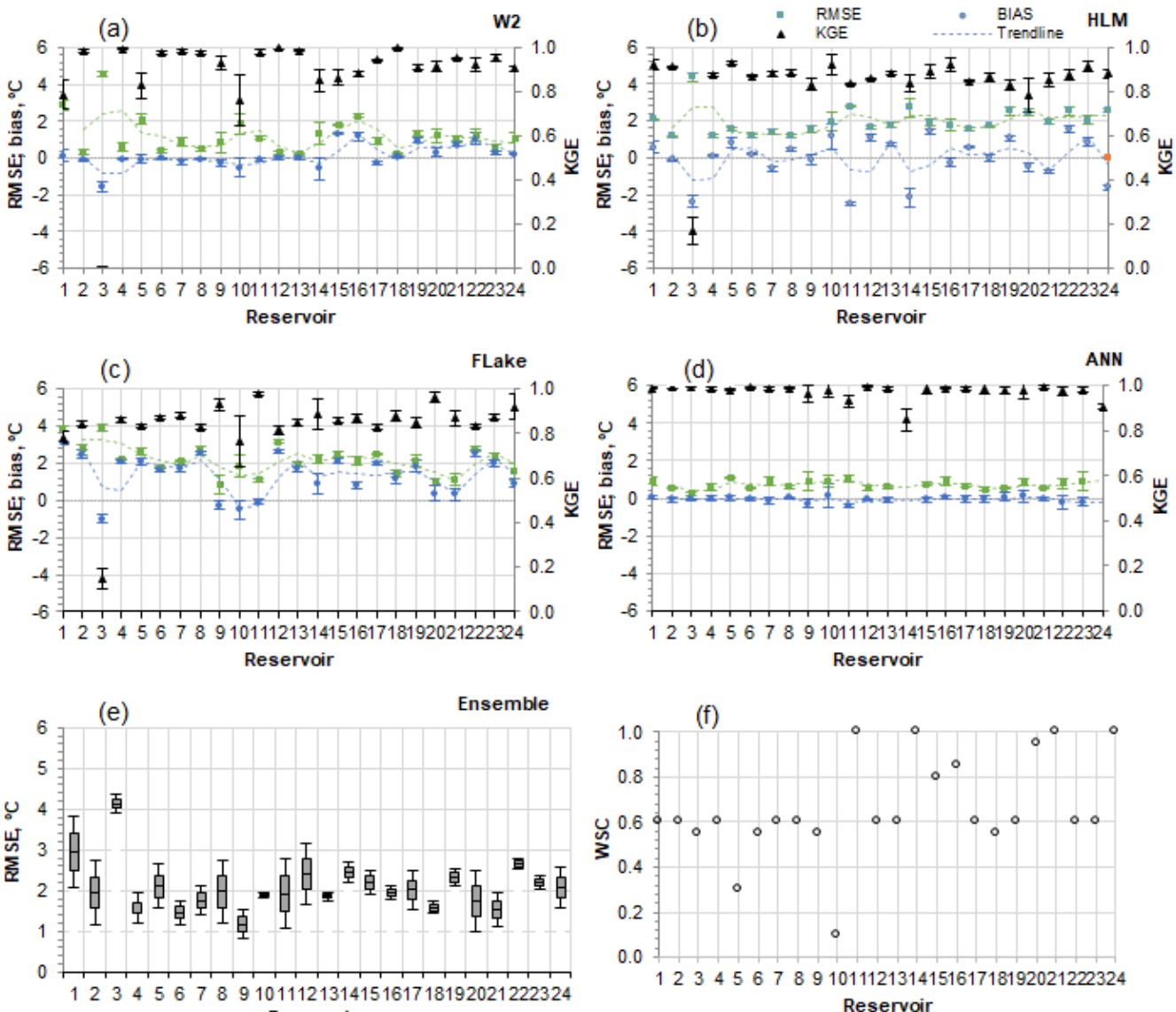

Figure 5. Evaluation of simulation bias, RMSE and KGE. 2-D baseline scenario (W2 hydrology -H) simulated SWT *versus*: a) the exclusion of inflows and outflows (W2 -H); b) HLM-H; c) Flake-H; and d) ANN-H. Models forced with hourly meteorology for the 24 waterbodies (2005-2008). The ensemble graphic, e), shows a box-plot of RMSE values (maximum, 75th percentile, median, 25th percentile, and minimum) considering the 1-D models results (2005-2008). In f) the wind sheltering coefficient considered during the calibration of the W2 hydrology scenario is presented.

Overall, the statistical comparison by Taylor diagrams (Fig. 6) suggests that FLake had a slightly better performance than HLM in simulating SWT. Noteworthy, the standard deviation of the simulations forced with hourly meteorology was consistently closer to the standard deviation of the baseline scenario (W2 hydrology-H) (Figs. 6c and d), showing the importance of meteorological data temporal resolution. ANN results were closer to the baseline scenario than the 2-D model (W2-H) regardless of the meteorological data discretization.

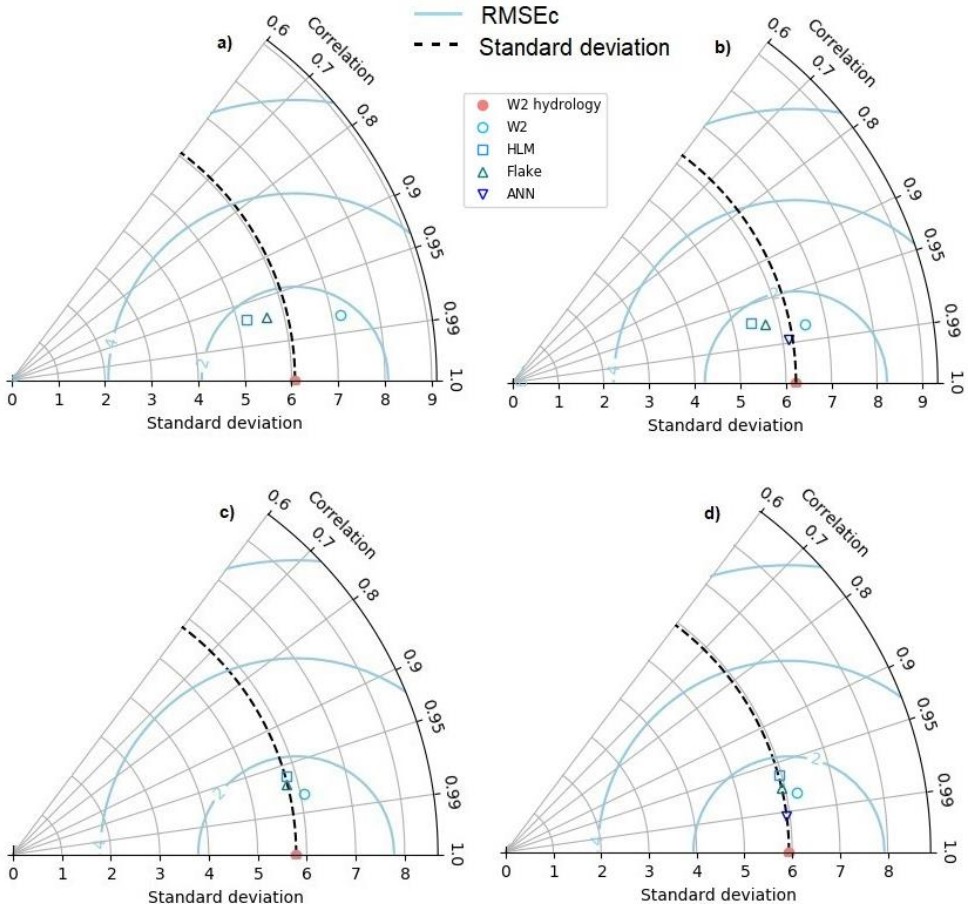

**Figure 6. Taylor diagrams showing standard deviation (ºC), RMSEc (ºC), and correlation of SWT for the baseline scenario (W2 hydrology) and for each other scenario (a) 1989 – 2008 (daily met.); b) 2005 – 2008 (daily met.); c) 1989 – 2008 (hourly met.); and d) 2005 – 2008 (hourly met.). Statistics are calculated over all 24 reservoirs/lakes for the 1989-2008 period, and over 22 reservoirs/lakes for the 2005-2008 period (Alqueva, R24, and Pedrógão, R14, reservoirs were not modeled with the ANN).**

### 4.2.2 Modeling computation time

The analysis of computation times was conducted through the comparison of the mean CPU time per time-step in the case of the 1-D models, with the mean CPU time per prediction sample obtained with the ANN, across all reservoirs. The results (Table 7) show, considering the hourly simulations that the prediction phase of the ANN is approximately 26 times faster than FLake, the fastest process-based 1-D model optimized for coupling with climate models. The HLM model code written in

Python is approximately 45 times slower than FLake, nevertheless it is important to mention that a Fortran implementation of Hostetler can be much faster as described by Thiery et al., (2016). In their work the Hostetler model was only 3.6 times slower

than FLake during the modeling of a deep lake (60 m deep). Table 7 also shows the significant difference in computational time between the 2-D model and all the other models.

It is important to mention that the performance of the models depends on the software implementation, therefore, the computation time values can vary significantly from the ones presented in this research.

**Table 7. Computation time for process-based physical models and for the ANNs prediction phase**

| | Daily meteorological forcing | | | | Programming language |
|---|---|---|---|---|---|
| **Model** | Number of layers | Number of time-steps | CPU time (s)/time-step | Total CPU time (s) | |
| W2 | *Vide* Table 4 | (1) | (1) | $3\,774 \pm 5\,818$ | Fortan 90/95 |
| HLM | $30.0 \pm 14.1$ | $6\,209 \pm 1\,786$ | $0.8 \pm 1.7 \times 10^{-2}$ | $9.5 \pm 5.0$ | Python 3.7 (Numpy 1.19.1) |
| Flake | 2 | $6\,209 \pm 1\,786$ | $1.8 \times 10^{-2} \pm 0.2 \times 10^{-2}$ | $0.2 \pm 3.7 \times 10^{-2}$ | Fortran 77 |
| **Model** | Number of training samples | Number of Predictions | CPU time (s)/number of prediction samples | Total CPU time (s) | **Programming language** |
| **ANN** | $3\,577 \pm 1185$ | $1\,192 \pm 395$ | $0.7 \times 10^{-3}$ | $6.9 \times 10^{-3} \pm 8.0 \times 10^{-4}$ | C++ and Python 3.7 |
| | Hourly meteorological forcing | | | | |
| **Model** | Number of layers | Number of time-steps | CPU time (s)/time-step | Total CPU time (s) | **Programming language** |
| W2 | *Vide* Table 4 | (1) | (1) | $4\,200 \pm 5\,848$ | Fortan 90/95 |
| HLM | $30.0 \pm 14.1$ | $149\,016 \pm 42\,866$ | $0.8 \pm 1.7 \times 10^{-2}$ | $163.0 \pm 89.2$ | Python 3.7 (Numpy 1.19.1) |
| Flake | 2 | $149\,016 \pm 42\,866$ | $1.9 \times 10^{-2} \pm 0.2 \times 10^{-2}$ | $3.0 \pm 0.9$ | Fortran 77 |
| **Model** | Number of training samples | Number of Predictions | CPU time (s)/number of prediction samples | Total CPU time (s) | **Programming language** |
| **ANN** | $117\,924 \pm 36\,785$ | $29\,481 \pm 9\,196$ | $0.7 \times 10^{-3}$ | $1.8 \times 10^{-2} \pm 3.2 \times 10^{-3}$ | C++ and Python 3.7 |

(1) The model dynamically computes a "stable" timestep with the autostepping algorithm (*vide* Cole and Wells, 2008)

## 4.3 The influence of reservoir inflows and level variations on SWT predictions

Additionally, to fully evaluate the influence of the inflows and level variations on SWT predictions in the reservoirs, and as a result, on surface latent and sensible heat fluxes, we considered the mean annual SWT results obtained with all models for six

reservoirs with SWT most sensitive to the exclusion of inflows/outflows. The reservoirs were chosen based on the six highest maximum RMSE values obtained between 2-D baseline scenario (W2 hydrology-H) SWT time series and SWT time series simulated with the exclusion of inflows and outflows (W2-H) (Fig. 7). Mean annual wind velocity, surface latent and sensible heat fluxes in these reservoirs are presented in (Figs. 8, 9 and 10), respectively. The results for the small Penide reservoir (R3), with a maximum depth of 9 m and an average volume of 0.11 hm$^3$, while revealing large errors in all model runs, also show

that these errors were significantly improved by the ANN. The HLM overperformed the FLake model in four of the six waterbodies (R1, R16, R5 and R22). Nevertheless, both 1-D models had an overall comparable performance. The ANN reduced significantly the annual maximum RMSE obtained for all reservoirs with all the models (Fig. 7).

The aggregated analysis of results presented in Figs. 8, 9 and 10, allows estimating the combined effect of the wind forcing and the influence of inflows and level variations in the surface heat fluxes. Separation of the wind effects from the mass budget variability is possible because the differences between W2 hydrology-H and W2-H scenarios describe only the combined influence of inflows and level variations on SWT, whereas the results obtained with the 1-D models describe the joint influence of the wind forcing and the influence of inflows and level variations (Figs. 9 and 10). The results obtained for reservoirs R1, R3, R5 and R10 show an appreciable effect on the surface heat fluxes caused by the neglect of inflows. As expected, the mean annual surface heat fluxes increased and decreased during the dry and wet seasons, respectively (Figs. 9 and 10). However, results obtained with the 1-D models, reveal a strong effect of the wind forcing across all reservoirs except reservoir R16. The differences in surface heat fluxes were as expected less pronounced in reservoir R16, due to the smaller difference between the wind forcing of the models (15%) (Fig. 8). Generally, the 1-D models overestimated the latent heat fluxes, in particular HLM, because FLake model results demonstrated a significant underestimation of SWT for reservoirs R1, R5, R10 and R22 as described by the corresponding maximum RMSE (Fig. 7). Accordingly, the mean annual sensible heat fluxes had a larger daily variability due to the need to balance the differences between air and water temperatures reaching, 21.09; SD $\pm$ 4.12 W/m$^2$ (Figure 10). The ANN reduced significantly the annual bias obtained for the surface heat fluxes for all reservoirs with all the models (Figs. 9 and 10). The only exception were the results obtained for R16, a run-of-the-river hydropower scheme, for which the 2-D modeling results were strongly affected by computation instability due to large inflow values. The training of the ANN partially reflected this instability into the final ANN structure causing a small overestimation of the surface heat fluxes during the dry season (Figs. 9 and 10).

Overall, the results show that the water-level variations are clearly related to surface-water temperature simulation bias; besides, the outflow (deep abstraction) reduces the volume of hypolimnion and increases the volume of the epilimnion (mixed layer) by lowering the thermocline. Herewith, water-level reduction increases the area-to-the-epilimnion volume ratio, which results in an increase in epilimnetic temperature (e.g., Carr et al., 2020). The hypolimnion water temperature (HWT) was generally higher in the W2 hydrology scenarios than in the W2 scenarios, due to the heat transported by interflow and underflow currents.

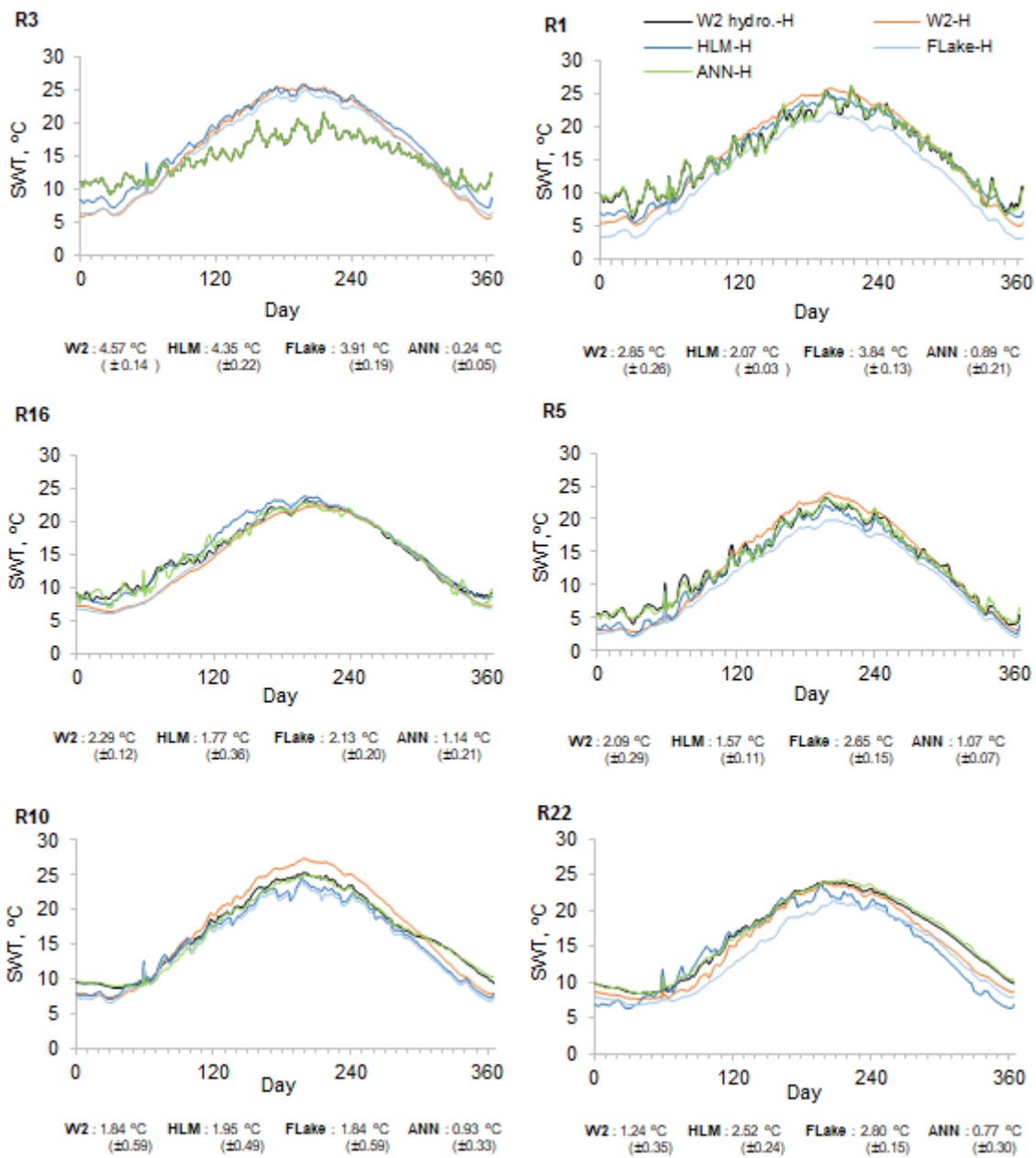

**Figure 7. Mean annual SWT values obtained with W2 hydrology-H (W2 hydro.-H), W2-H, HLM-H, FLake-H and ANN-H scenarios considering hourly meteorology (2005-2008). Annual Maximum RMSE between W2 hydrology-H (W2 hydro.-H) and the other scenarios SWT results (Graphics are ordered from the highest to the lowest RMSE values obtained for W2-H scenarios)**

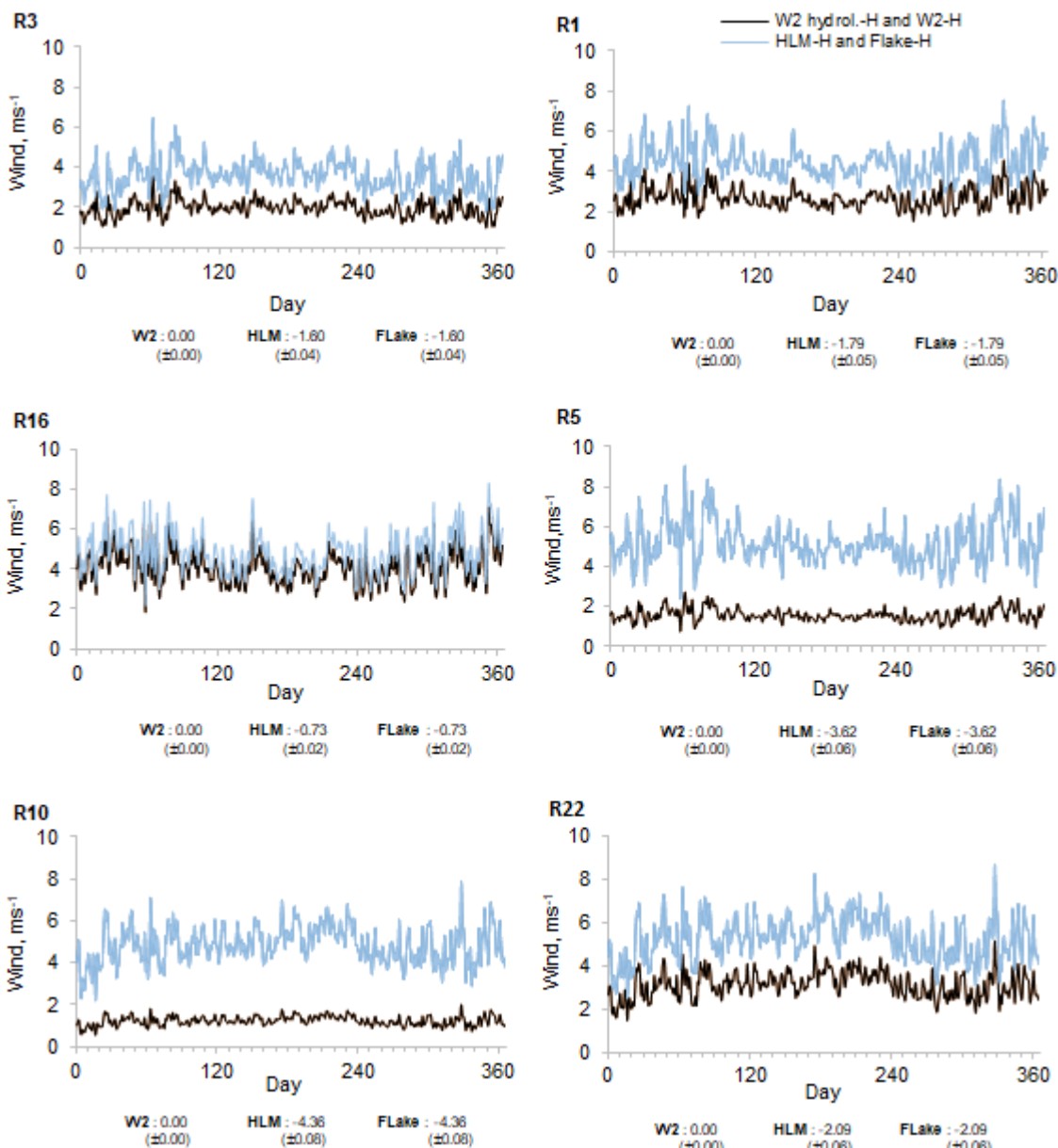

**Figure 8.** Mean annual wind-velocity values obtained with W2 hydrology-H (W2 hydro.-H), W2-H, (accounting for the wind-sheltering effect), HLM-H and FLake-H scenarios taking into consideration hourly meteorology (2005-2008). Bias between W2 hydrology-H and the other scenarios' mean wind-velocity values


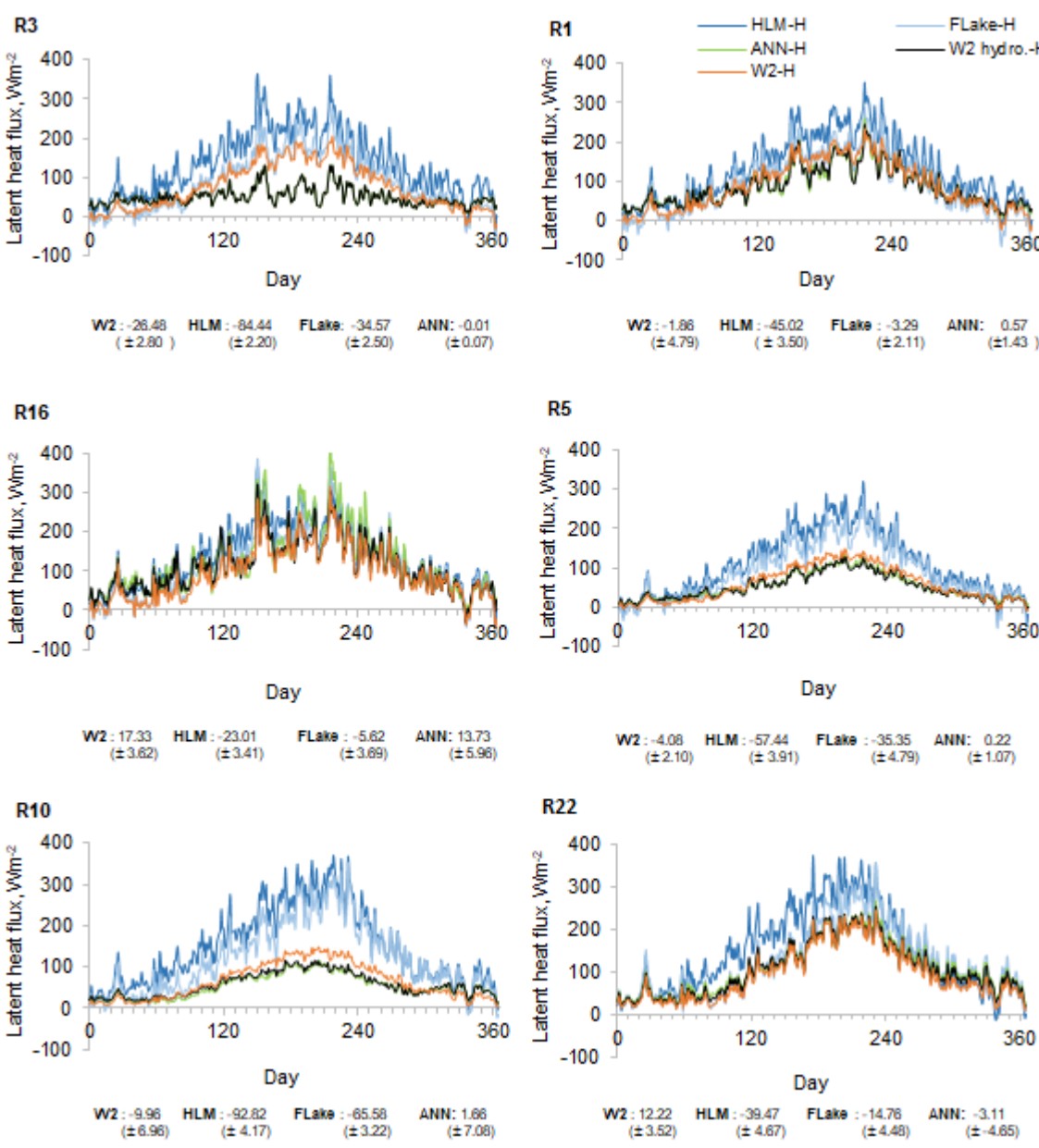

**Figure 9. Mean annual latent heat values obtained with W2 hydrology (W2 hydro.-H), W2 scenarios (W2-H), HLM and FLake, considering hourly meteorology (2005-2008). Bias between W2 hydrology (W2 hydro.-H) and the other models SWT results**

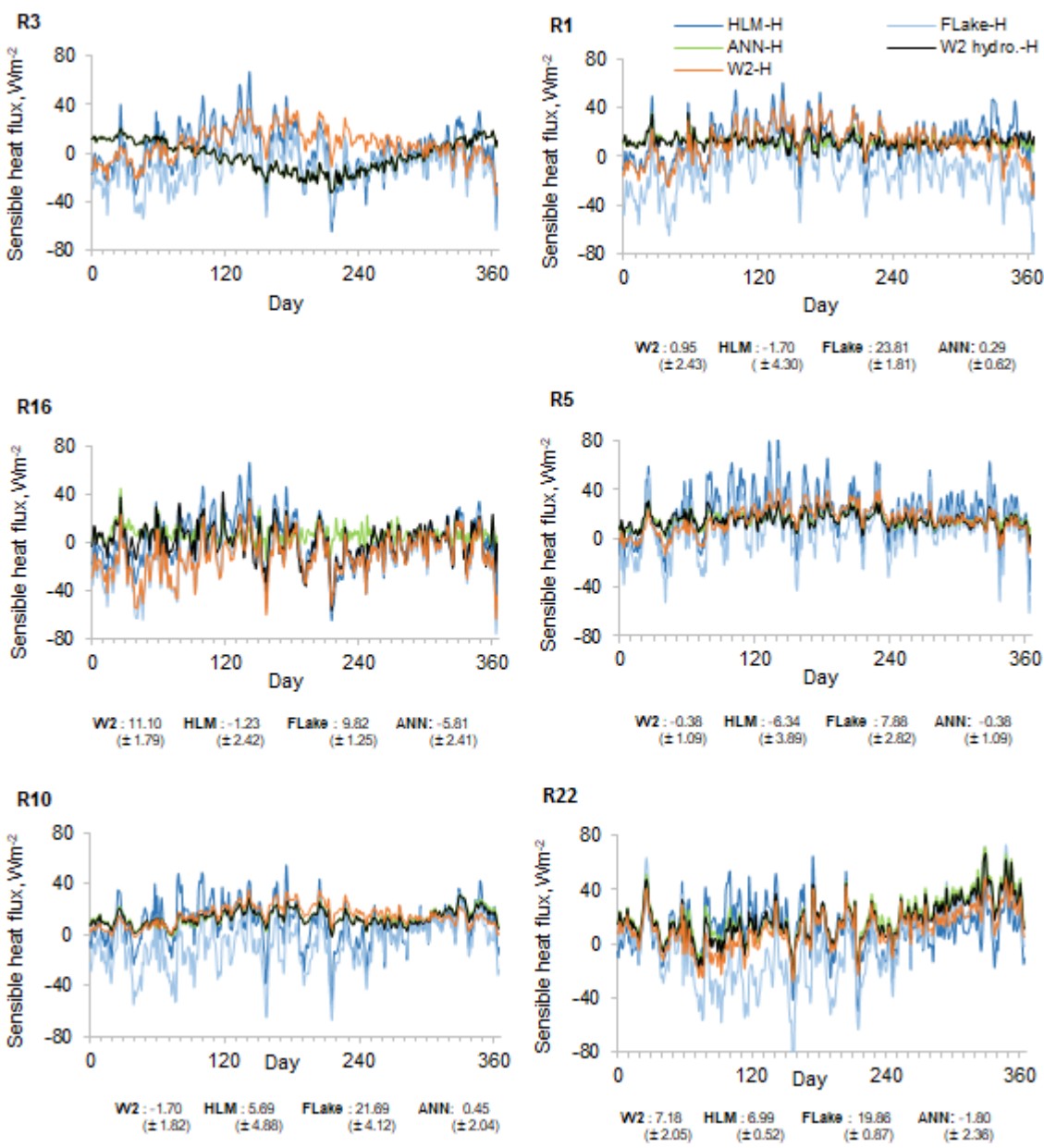

**Figure 10. Mean annual sensible heat values obtained with W2 hydrology (W2 hydro.-H), W2 scenarios (W2-H), HLM and FLake, considering hourly meteorology (2005-2008). Bias between W2 hydrology (W2 hydro.-H) and the other models SWT results**

The differences between W2 hydrology and W2 scenarios describe quite well the combined influence of inflows and level variations in SWT evolution, which can be parametrized using the WRT. The results obtained for both scenarios reveal a significant logarithmic correlation (Eq. 10) between the RMSE of SWT from the two scenarios and WRT (Figure 11a).

$$RMSE = -0.36 \ln(WRT) + 2.73, \qquad R^2 = 0.88; MAE = 0.27\ °C \qquad\qquad (11)$$

, with RMSE and WRT expressed in °C and in days, respectively.

The results additionally show that the computed SWT values in reservoirs with a residence time shorter than 100 days may have large errors if simulated without inflows/outflows (Figure 11).

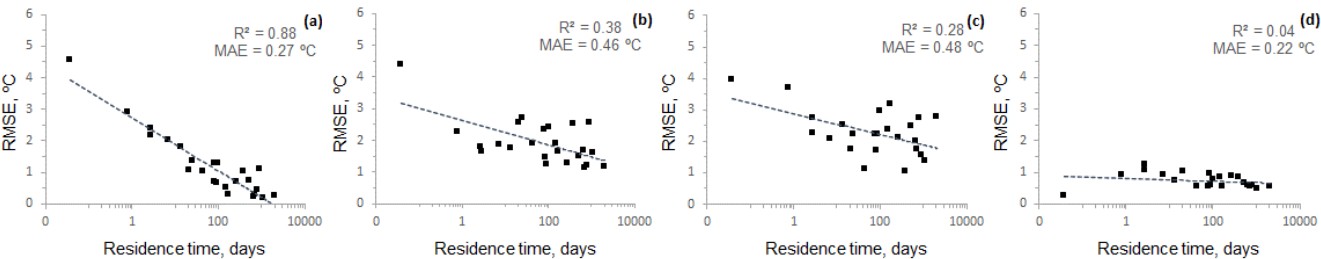

**Figure 11. RMSE as a function of WRT. Between the W2 hydrology-H scenario and simulated SWT with: a) the exclusion of inflows and outflows (W2-H, 1989-2008); b) HLM-H (1989-2008); c) FLake-H (1989-2008); and d) ANN-H (2005-2008), driven with hourly meteorology.**

**5 Discussion and conclusions**

The thermodynamics of natural and artificial lakes are similar. Nevertheless, the evolution of SWT in lakes and reservoirs
differs substantially as a result of heat advection by inflows, outflows and, to a lesser extent, due to water level variations. Evaluation of differences between thermal regimes of lakes and reservoirs from observational data is limited by the availability of comparable waterbodies. The model-based approach used in the present study provides an effective alternative, complementary to the studies evaluating the thermal structure differences between lakes (similar to a seepage lake) and reservoirs across the latitudinal gradient (e.g., Doubek and Carey, 2017; Hayes et al., 2017). We show that, for the same
morphometry and under the Mediterranean climatic conditions, the SWT in reservoirs (approximately 46 %) is higher than the SWT in lakes (similar to a seepage lake). The results also suggest that the SWT predictions can be significantly affected by the water surface level variations. Nevertheless, in the present study only the combined effect of advection/level variation was evaluated, and the individual effect of level variations was not correlated with the SWT simulation errors. Therefore, the partial contribution of this variable to SWT was not fully explored and requires a future in-depth analysis.
One of the main novel aspects of the study lies in the fact that computationally efficient models (1-D and ANN) are compared against a baseline target instead of among themselves. Additionally, the study relies on the analysis of a large number of waterbodies and simulations conducted over a several decades long period. The methodological approach exposed the strengths

and weaknesses associated with the simulation of the SWT of reservoirs by both process-based physical and data-driven models. We demonstrated that inflows and outflows have a relevant effect on the evolution of SWT, with broader implications in the quality of GCMs and RCMs used in numerical weather prediction and climate modeling. It was also shown that there are other factors besides inflows and outflows that affect SWT. Examples are the wind forcing, the temporal sampling of the meteorological forcing data and the simplification of processes for quantifying turbulent energy flows. The low computational costs of 1-D process-based models in particular of the FLake model is the decisive factor for their integration in numerous GCMs and RCMs. Indeed, 1-D models such as FLake and HLM present a particularly good alternative to model reservoirs with missing field data and external parameters. Overall, Hostetler and FLake models demonstrated a reasonable performance, the latter being slightly better in modeling SWTs. Nevertheless, the results highlight that their SWT predictions can diverge significantly from observed values unless advective heat transport by in- and outflows and water level variations are integrated in the models. As an alternative to process-based models, an improvement can be achieved both in accuracy and computational requirements by using data-driven models. The ANN approach demonstrated a remarkably good performance by reducing the average value of RMSE of hourly simulations by at least 64% and running 26 times faster than FLake model. Nevertheless, there are two important limitations to the implementation of ANNs in GCM or RCM contexts. The first is the need for sufficient amount of accurate observational data to train the model; the second is the availability of river inflow temperatures. Both are still scarce, but their availability is rapidly increasing due to recent developments in remote sensing.

The present results suggest that reservoirs with a WRT shorter than 100 days, if simulated without representation of inflows and outflows, tend to exhibit an important deviation in the computed SWT values regardless of their morphological characteristics. Neglecting inflows and outflows while modeling these waterbodies may cause an overestimation of the turbulent energy fluxes, which can produce spurious local instabilities if surface water temperatures are higher than mean air temperatures.

Incorporation of inflows and outflows in 1-D models for regional and global climate simulations will decrease computation efficiency and add an additional layer of uncertainty in the modeling of systems whose real nature is three-dimensional. The data-driven model considered in this study outperformed process-based physical models in computation time and in accuracy, being capable of accounting for the influence of inflows and outflows. In the context of waterbody simulation within numerical weather prediction and climate models, the use of data-driven approaches to complement their process-based counterparts may be highly efficient when data necessary to train the models is available. Given the growing capabilities and increasingly common use of remote sensing data acquisition techniques, the possibility of improving the performance of GCMs and RCMs through the enhanced modeling of waterbody-atmosphere turbulent heat exchanges is promising.

**Code availability**

The exact version of the models source code is archived on Zenodo at http://doi.org/10.5281/zenodo.4803480 (Almeida, 2021a). The current version of the open-source CE-QUAL-W2 model (version 3.6) used in this study, is also available from

the project website (http://www.ce.pdx.edu/w2/). FLake (version 1.0) is freely available under the terms of the GNU Lesser General Public License (http://www.gnu.org/licenses/lgpl. html). The model source code, a windows executable, as well as a comprehensive model description are freely available from the official FLake website (http://www.lakemodel.net). For completeness, the windows pre-compiled version of FLake as used in the present calculations is also archived on Zenodo (Almeida, 2021a). The open-source Hostetler model source code is also available from the repository. The Python library used

to construct the ANN, NeuPy version 0.8.2, is available from the NeuPy website (http://neupy.com/pages/home.html) under the terms of the MIT License and the ANN source code and scripts used to train the model are archived on Zenodo (Almeida, 2021a).

### Data availability

Input files needed to run the models and the hydrometric, water quality and meteorological data sets used to force and validate

each model, are freely available and are archived on Zenodo at http://doi.org/10.5281/zenodo.4756312 (Almeida, 2021b).

### Author contribution

MA conceived the study, produced the code of Hostetler model and performed the simulations. YS developed and optimized the ANN, while GK provided support to FLake model simulations. PS and RC provided the meteorological data sets. JP supported code development. PC, AC and RR contributed to the study design and to the results analysis. All authors contributed

to the discussion and manuscript revision.

### Competing interests

The authors declare that they have no conflict of interest.

### Acknowledgments

The authors thank the Portuguese Environmental Agency for providing the hydrometric and water quality data sets that were

used in this study. Sincere thanks are given to one anonymous reviewer for the invaluable and constructive comments and suggestions for improving the paper quality.

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
