# Peer review of "Modeling reservoir surface temperatures for regional and global climate models: a multi-model study on the inflow and level variation effects"

_Geoscientific Model Development, 2021_

## Author Response (AR1)

**FINAL-RESPONSE FORM**

Dear Editor,

We appreciate the work of the referees in helping us to improve the manuscript. Many thanks for the guidelines and constructive comments to our manuscript. We now present our revised manuscript and our replies to the reviewers' comments and suggestions.

Please find below the point-by-point responses to the reviewers' comments and the new manuscript with (and without) highlighted changes. We hope the revised manuscript is now acceptable for publication in Geoscientific Model Development. All authors agree on the current form of the manuscript.

Dr. Manuel Almeida, on behalf of the authors.

**Anonymous Referee 1:** This manuscript investigates the role of inflows and outflows as well as water level variations in reservoirs on the accuracy of surface water temperature predictions by models. Its relevance and main motivation is to improve representation of surface-water - atmosphere interactions in numerical weather prediction, but it also has important implications for limnology. I think the topic is novel and important, as the effect of heat flux through water advection is generally neglected both in NWP and limnological investigations on, for instance, climate change. Reservoirs are seldom specifically studied, yet they have considerably different characteristics compared to lakes, especially their riverine character and the importance of hydrology. The study showed that inflows and outflows indeed have an effect on surface water temperature, particularly in reservoirs with a retention time below 100 days, which should accordingly affect a very large number of reservoirs, which generally have shorter retention times than lakes. A particular strength of the study is its ensemble approach of using one and two-dimensional process-based models of varying complexity as well as a machine learning model, applied to a considerable number (24) of Portugese reservoirs spanning a range of sizes and retention times. This enabled a detailed a robust analysis of the influence of retention time. The model intercomparison is useful in numerical weather prediction because the necessarily high computation efficiency required means that model complexity and input data resolution need to be examined and optimized. I think this paper will make a valuable contribution to the field.

**RESPONSE:** Thank you for the time taken and for your thoughtful comments and suggestions for improving our manuscript. To facilitate the work of the reviewers and editor, when responding we will refer to the original manuscript and indicate the line that was modified in each case.

**COMMENT:** I have a couple of issues with the study. Firstly I would avoid the comparison of reservoirs with lakes. Lakes have inflows and outflows just as reservoirs do, and some lakes also have very short retention times. Furthermore, only reservoirs and no lakes were investigated in the study. An alternative designation for the baseline scenarios may be just W2 instead of W2-Lake, and W2-hydrology instead of W2-reservoir, but this is just an idea.

**RESPONSE:** Thank you. It is a good idea. As suggested, we have made changes regarding the description of the scenarios, and we have also changed the designation of the scenarios. Furthermore, we have replaced the term "lake" with "lake (similar to a seepage lake)".

**COMMENT:** Secondly, the main difference between reservoirs and lakes in my opinion is not the presence/absence of inflows and outflows (though reservoirs are

typically dominated by hydrology as riverine systems), but the fact that most reservoirs have deep water outlets, whereas lakes discharge from the surface. This can have profound effects on the heat budget in stratified waters because of potentially large differences in surface (e.g. 25 degrees C) and bottom water temperature (eg 10 degrees C or potentially lower). Deep water withdrawal tends to shrink the hypolimnion, expand the surface mixed layer, decrease vertical temperature gradients, shorten stratification duration, and increase the heat content of reservoirs in comparison to lakes with surface water outlets. This aspect was not addressed in the paper but would significantly influence surface water temperature and I recommend that the authors provide information on whether/how this was accounted for, how the outflows were represented in the model, and what its potential effect on the results would be.

**RESPONSE:** Thank you for your comment. We agree that deep-water withdrawal can have a significant effect on surface-water temperature and that this is one of the main differences between lakes and reservoirs. The 2-D model considered in the definition of the baseline scenario includes a selective withdrawal algorithm that calculates a withdrawal zone based on outflow, outlet geometry and upstream density gradients (Cole and Wells, 2008). Hence, the effect of deep withdrawal was accounted for.

During the development of this study, we have included 24 reservoirs with different water-residence time and morphological characteristics (volume, depth, surface area). We have taken this large number of waterbodies into consideration, including run-of-river hydropower schemes, to also evaluate the effect of deep-*versus* shallow-water withdrawal outlets. In fact, we have evaluated the relationship between the water level, mixed-layer depth, the thermocline depth and the surface-water temperature variation for each reservoir, between the W2 reservoir and the W2 lake scenarios.

These were the conclusions regarding this specific analysis: the water-level variation is clearly related to surface-water temperature simulation bias, and we show that the outflow (deep abstraction) reduces the volume of hypolimnion and increases the volume of the epilimnion and of the mixed-layer depth (i.e., the thermocline depth increases). The water-level reduction increases the area-to-the-epilimnion volume ratio, which results in an increase in epilimnetic temperature (e.g., Carr et al. 2020). The hypolimnion water temperature (HWT) was generally higher in the reservoirs than in lakes, due to the heat transported by interflow and underflow currents.

These conclusions have already been well accepted by the scientific community. Therefore, after considering the length of the manuscript versus the scientific benefit of the inclusion of this specific analysis, we have decided to exclude this part from the final version of the manuscript. While quantification of the individual contribution of the outflow-withdrawal depth would be relevant to the study, we only focused on the integrated effect of advection/level variation in this study.

Our revised manuscript will include a description of the outflow representation in the 2-D model, and the following sentences will be included:

Line 205 - the following sentence was modified:

"Surface-heat fluxes, in particular the evaporation rate, are also affected by advection due to inflows and outflows (e.g., deep-water withdrawal) and by water level (WL) fluctuations (Rimmer et al., 2011; Friedrich et al., 2018)."

Line 205 - the following sentences were included: "Outlet geometry, outflows and in- pool densities are the input to the selective withdrawal algorithm that calculates vertical withdrawal zone limits. Among the two model options of the withdrawal—line sinks, which are wide in relation to dam width (> 1/10) and point sinks, which are narrow in relation to dam width (< 1/10)— only point sinks were considered. The point-sink approximation assumes the flow is radial, both longitudinally and vertically (Cole and Wells, 2008). Therefore, for the outflow structure definition, the centerline elevation of the structure was included in the model (Table 4). Additionally, as suggested by Cole and Wells (2008), the algorithm was allowed to retrieve water from the top elevation of the computational grid."

Line 447 - the following sentence was included:

"Overall, the results show that the water-level variations are clearly related to surfacewater temperature simulation bias; besides, the outflow (deep abstraction) reduces the volume of hypolimnion and increases the volume of the epilimnion (mixed layer) by lowering the thermocline. Herewith, water-level reduction increases the area-tothe-epilimnion volume ratio, which results in an increase in epilimnetic temperature (e.g., Carr et al., 2020). The hypolimnion water temperature (HWT) was generally higher in the W2 hydrology scenarios than in the W2 scenarios, due to the heat transported by interflow and underflow currents."

**COMMENT:** Thirdly, I found the description of the scenarios somewhat confusing and I have some questions. Could you please describe the rationale behind the scenarios in more detail and perhaps early on (e.g. the baseline scenario is actually 4 scenarios - hourly, and daily forcing, with and without hydrology). Please also clearly indicate whether the 1D model scenarios considered inflows and outflows or not (I believe it was mentioned in the introduction but it should be clear in the methods too). Consider adding a table describing the different scenarios and comparisons. You may also consider removing Table 2 as its content can easily be summarized in the text.

**RESPONSE:** Thank you for your comment. We agree with the reviewer. Therefore, Table 2, previously included in the methods section, was replaced with a table describing the different scenarios, which clarifies the rationale behind the scenarios.

Line 159 - Table 2 was summarized in the text by the following sentence: "Meteorological datasets considered in the modeling process included: air temperature (°C); relative humidity (%); wind velocity (m/s); wind direction (rad); cloud fraction (0 to 10) and shortwave-solar radiation (W/m2). These datasets were considered in all models with the following exceptions: wind direction is not considered for 1-D models forcing; the ANN modeling relays in the air-temperature, relative-humidity and wind-velocity datasets only."

Line 138 - The following sentence was included: "Table 2 shows a full description of the scenarios considered in the development of this study."

Line 161 - Table 2 was replaced.

**COMMENT:** Finally, the model errors were assessed against the observed data (correct in my opinion) but in other analyses model errors were assessed against the baseline scenario W2-Reservoir, as far as I can ascertain. This can be problematic because each model has its own characteristics and does not per se represent the "truth", especially if some parameters are highly calibrated as is the case of the baseline scenarios here. For instance, if the extinction coefficient was calibrated in the baseline scenario with the 2-d model but not in the 1-d models, or if different parameter values were used, are the results comparable and how meaningful are the model errors? I think it may make more sense to calculate the model error always against the observed values, and then to assess the relative changes in error in the different scenarios. E.g. "not accounting for inflows increased the model error by X". If the model error is used mainly as an analytical variable, please describe the rationale here clearly.

**RESPONSE:** Thank you for your comment. Indeed, in the first part of the manuscript results, the models' errors were assessed against the observed temperature values (profiles and surface-water temperature values) taking into consideration a period of 20 years.

Note that the error obtained by comparing the models' results with the observed values also includes a partial error that can be associated with the uncertainty of the model forcing data from the downscaled dataset of ERA-interim reanalysis. Since the 1-D simulations do not include inflow/outflow and water-level variations, it was not possible to assess if the model errors were linked to the neglect of inflows and outflows or caused by the forcing errors.

Therefore, at the second stage of the model-performance analysis, model errors were assessed against the baseline scenario ("W2 hydrology"). The reviewer is right, at this point the error should be interpreted as an analytical variable used to answer the following questions:

- a) What is the exact variation that is associated with the neglect of inflows and outflows? This also enabled the definition of the 100-days retention time threshold.
- b) How well can ANN simulate the evolution of a reservoir SWT?
- c) How far apart is the performance of this simplified 1-D model from a state-ofthe-art calibrated 2-D model that includes the parametrization of inflows/outflows and water-level variation? What is the partial contribution of the neglect of inflows and outflows versus the effect of wind sheltering over the models' meteorological forcing to the final error value;
- d) Can we identify differences in the conceptualization of important physical processes (e.g., differences in the conceptualization of diurnal variations of SWT between 1-D and the 2-D models), through the consideration of daily versus hourly meteorological forcing?

Regarding the baseline scenario "quality", it is important to mention the following:

- a) One of the two calibration parameters—wind-sheltering coefficient (WSC)—is directly linked to the effect of the watershed morphology on wind forcing and accounts, therefore, for potential uncertainties in meteorological input (Cole and Wells, 2008).
- b) The second calibration parameter was the water-extinction coefficient. These two parameters are commonly associated with a high degree of uncertainty in regional and global climate models. It is noteworthy that the WSC had the strongest effect on temperature during calibration.
- c) The CE-QUAL-W2 model has been extensively used in the last 20 years to reproduce water temperatures in Portuguese reservoirs, as well as in over 400 waterbodies worldwide under a wide variety of external conditions. The model demonstrated remarkably accurate temperature predictions provided the accurate geometry and boundary conditions are available (*vide* Cole and Wells, 2008).

Therefore, we believe that the 2-D model, after calibration, represents a reliable benchmark for testing the representativeness of simplified 1-D models for reservoir parameterization in regional and global models.

To resolve the reviewer's concerns and to clarify our approach, the following changes have been proposed to the revised version:

Line 155: the paragraph has been added: "The baseline scenarios (W2 hydrology) were defined to address the following questions:

- e) How large is the uncertainty associated with the neglect of inflows and outflows?
- f) How adequate is the performance of simplified 1-D models compared with the state-of-the-art calibrated 2-D model, including parametrization of inflows/outflows and WL variation? What is the relative contribution to the final model error of the in- and outflow neglect vs. neglect of the wind sheltering in meteorological forcing?
- g) Can we identify conceptual differences in representation of the fundamental physical processes (such as differences in the conceptualization of diurnal variations of SWT) by 1-D and 2-D models through the comparison of outputs from daily versus hourly forcing?
- h) How well can ANN simulate the evolution of a reservoir SWT?

The reliability of the baseline scenarios (W2 hydrology) for representation of the reservoir thermal regime has been demonstrated by the model calibration results and is supported by the outcomes of a large number of successful model applications worldwide (*vide* Cole and Wells, 2008).

Using 2-D modeling results as a baseline "benchmark" scenario for validating 1-D models allows the isolation of the errors associated with the quality of meteorological forcing and observed data (e.g., water-temperature data sets) while providing the continuity usually unavailable from observational datasets. Hence, the error obtained when comparing 1-D versus 2-D model results is to be regarded as an analytical variable, encapsulating differences among the different scenarios and not the conventional model error (model output *versus* observed data)."

Minor points

Abstract

**COMMENT:** L 20-21: "Our results highlight that surface water temperatures in reservoirs differ significantly from those found in lakes" is slightly misleading as you did not investigate any lakes. I suggest you reword the sentence to reflect your actual findings, and from that make inferences about lakes if appropriate.

**RESPONSE:** Thank you for your comment. We agree, the sentence has been changed.

Line 20 -21 was replaced with the following sentence: "Our results highlight the need to include anthropogenic inflow and outflow controls in regional and global climate models"

Introduction

**COMMENT:** The aims don't seem to include a comparison of the effect of in and outflows.

**RESPONSE:** Thank you for your comment. The reviewer is right: the main objective of the study was to evaluate the combined effect of in- and outflows without their separation.

**COMMENT:** L 89 – 91: This sentence is a little confusing. Try: Maximum daily mean air temperature ranges from 13 °C in the central highlands to 25 °C in the southeast region. The minimum daily mean air temperature ranges from 5 °C in the northern and central regions to 18 °C in the south.

**RESPONSE:** Thank you for your comment. We agree, the sentence has been replaced.

**COMMENT:** L118 – 120: Did you use single linear regressions between air and water temperature? There is typically a seasonal hysteresis in the air-water temperatures that varies in strength depending on some factors (eg Q), so that you would probably get a better estimate of inflow water temperature using a model like air2stream (https://github.com/marcotoffolon/air2stream, see references here). I am not suggesting you do this and repeat all the simulations and analyses, but could you please comment on this effect?

**RESPONSE:** Thank you for the comment and for suggesting an alternative modeling approach. We agree with the reviewer: rivers, reservoirs and lakes reveal thermal inertia with regard to air temperature. During the model calibration process, we have tested some different approaches:

- i) Linear regressions between air and water temperature, taking into consideration full datasets, and also considering seasonal datasets;
- ii) A non-linear regression model (Mohseni O. et al., 1998);
- iii) An equation developed by Stefan and Preud'homme (1992), which is used by the SWAT watershed model and considers air temperature, a time lag between air and water temperature and the mean annual flow.

We are aware of the importance of the time lag between air and water temperature and its relation to the strength of in- and outflow. While the approximation described in (iii) is the most advanced among the three approaches above, the linear-regression (i) equations produced the best results based on monthly inflow values. We have not evaluated this issue further because we were satisfied with the results. Nevertheless, we agree that the results could have been improved if a model like air2stream was used. Therefore, we have added the following note in lines 136-139 of the manuscript:

"A deeper insight into the relationship between the air and surface temperatures may be obtained by application of more detailed semi-stochastic models (Toffolon and Piccolroaz, 2015), while the effects of the reservoir volume (depth) and the flow would require specific attention in this case (Calamita et al. 2021). "

**COMMENT:** L117: Could you please describe what the baseline scenario is? It would be helpful to explain the rationale of the scenarios early on and in a little more detail.

**RESPONSE:** Thank you for your comment. We agree with the reviewer. In order to address this problem, we have placed section 3.1 Forcing and calibration data after section 3.2 Models/scenarios. Thus, the scenarios' rationale appears before section 3.1

Line 117 was replaced with the following sentence: "The W2 hydrology scenario was forced..."

**COMMENT:** L131 – 134: why did you not use the solar radiation and cloud cover from ERA-Interim?

**RESPONSE:** Thank you for pointing this out. In an initial stage of the development of this study, we did not have access to the solar-radiation and cloud-cover datasets obtained from the Weather Research and Forecasting (WRF) model for Portugal. Therefore, to address this limitation we had two alternatives:

- i) To include ERA-Interim datasets for cloud cover and solar radiation;
- ii) To consider mean monthly cloud cover values observed in the closest meteorological station and apply the cloud cover reduction of clear sky solar radiation described by Wunderlich (1972), which is still included in the most recent version of the CE-QUAL-W2 model parameterization (Wells, 2021) and calculate the clear sky radiation with a suitable algorithm (e. g. Thackston and Parker, 1971).

We chose the second option, because:

 We could not find validation studies of ERA-Interim cloud cover datasets for Portugal, and it is well known that cloud cover in ERA-Interim is underestimated by at least 10% for all regions except the poles (Free et al., 2016, Stengel et al., 2018). This underestimation implies an overestimation of shortwave incident radiation, which is more significant in summer (Free et al., 2016, Zhang et al., 2020);

- ii) Cloud-cover datasets from reanalyses are obtained with physical model parameterizations, therefore they are also subject to error (Free, et al., 2016);
- iii) The spatial resolution of the ERA-Interim data set is approximately 80 km. This is an important limitation when taking into consideration 2-D model spatial resolution and above all, the known difficulties in modeling cloud formation (Jakob 1999, Bedacht et al. 2007, Clark and Walsh 2010, Wu et al. 2012);
- iv) We have always obtained good results with the solar radiation data sets obtained with the EPA method (Thackston and Parker, 1971), with the cloud cover datasets derived from mean monthly values described in the climatological normal of Portugal (1951-1980) and with the Wunderlich (1972) algorithm. Unfortunately, all projects were only written in Portuguese. However, they are available in the following address: <a href="https://apambiente.pt/agua/modelacao-da-qualidade-da-agua-em-albufeiras-de-aguas-publicas">https://apambiente.pt/agua/modelacao-da-qualidade-da-agua-em-albufeiras-de-aguas-publicas</a>

Considering the known difficulties in obtaining high-resolution and high-quality cloud cover observations, we think that the question from the reviewer is quite relevant. In the near future we will assess the differences between these two options, namely, their effect in the forcing of water-quality models.

**COMMENT:** Section 3.2: could you please describe how you parameterized the basin morphology for W2 simulations, especially the data sources and horizontal and vertical grid resolution, and outlet heights?

**RESPONSE:** Thank you for your comment. We have used 1:25000 charts to define the reservoirs' bathymetry. This information was included in the revised version of the manuscript on line 213:

"The reservoirs' bathymetry was defined from 1:25000 topographic charts of the watersheds. Hence, each reservoir computational grid is described by a specific number of branches, segments, and layers (Table 4)."

Additionally, we have included Table 4, which describes the computational grid dimensions of each reservoir (number of branches, layers, segments) and the outflow-centerline elevation regarding each outflow structure.

**COMMENT:** L 143-145: Please describe the calibration procedure. Which parameters were calibrated in which ranges? Was calibration automatic? Which extinction values did you use for the uncalibrated models?

**RESPONSE:** This comment is much appreciated. Details on the calibration procedure were included in the revised version of the manuscript. As described earlier in this text, we have calibrated two parameters (wind sheltering and water-extinction coefficient). The 2-D models were calibrated by comparing the results with observed data, without applying automatic adjustment algorithms. Due to the difficulties encountered in characterizing reservoir water transparency with observed data, all 1-D simulations were performed with a constant extinction coefficient value of 0.45 m-1. This extinction coefficient is the reference value considered by Cole and Wells (2008) for the model CE-QUAL-W2 when only water temperature is simulated, 0.45. According to the eutrophication criteria defined by the OCDE (OCDE, 1982), this magnitude of water transparency is associated with eutrophic unstable systems, which are becoming common in Portugal. This value is also close to the mean value obtained from observed Secchi disk data available for four reservoirs: Bouçã (R10), Crestuma-Lever (R16), Cabril (R22) and Castelo do Bode (R23), 0.37.

Line 145: The following sentence was included in the manuscript:

"After each model run, results were compared with the observed data sets and if needed the calibration parameters were retuned manually. The wind-sheltering coefficient (WSC) and the extinction coefficient for water were the only parameters modified at each model run. These parameters varied in the range from 0.1 to 1.0 and from 0.25 to 1.0, respectively. Data on the mean water-extinction coefficient was available for four reservoirs: Bouçã ( $\mu$ =0.27;  $\sigma$ =0.05), Crestuma-Lever ( $\mu$ =0.67;  $\sigma$ =0.15) - 0.67, Cabril ( $\mu$ =0.27;  $\sigma$ =0.05) and Castelo do Bode ( $\mu$ =0.26;  $\sigma$ =0.05), therefore they were not calibrated. All 1-D simulations were performed with a constant water-extinction coefficient value of 0.45, corresponding to the reference value suggested by Cole and Wells (2008). According to the eutrophication criteria defined by the OCDE (OCDE, 1982), this value of water transparency is associated with eutrophic unstable systems and is also close to the mean value of 0.37 obtained from the four reservoirs listed above."

**COMMENT:** L 174: Change to "Two different temporal resolutions ..."

**RESPONSE:** Thank you for pointing this out. This sentence was changed.

**COMMENT:** L 212: the sentence doesn't convey what the Ultimate algorithm is used for. I suggest to reword it.

**RESPONSE:** Thank you for your pointing this out.

Line 212 – The sentence was changed to: "The Ultimate algorithm was considered as the solution for the numerical transport for temperature and constituents (Cole and Wells, 2008)."

**Results:**

**COMMENT:** L 293-296: The parameter ranges appear to be quite big and the calibration resulted in especially low values of the wind factor and extinction. The average extinction value of 0.38 1/m is typical of reasonably deep oligotrophic lakes but several reservoirs are quite shallow and I would expect some considerably higher values. I am wondering if the low wind factors were compensating for the low extinction values? Were these calibrated values used for the other models, or which parameterisations were used?

**RESPONSE:** Thank you for this question. The reviewer is right, there are some low values of the wind-sheltering coefficient (WSC). In some cases, the wind velocity had to be considerably reduced. According to Cole and Wells, 2008, the wind-sheltering coefficient [WSC] has the most effect on temperature during calibration and should be adjusted first. Then, if needed, all the other calibration parameters can be adjusted, including the extinction coefficient. We have always followed this recommendation, which in our perspective prevents the unbalancing of the calibration process. Furthermore, the modeling of such a long time period (20 years) enabled the evaluation of the model's response under different forcing conditions, namely the wind stress. This fact per se, reduces the probability of validating an unbalanced model.

**COMMENT:** L302: change 'major' to 'high' or 'highest'.

**RESPONSE:** Thank you for your comment. The sentence was changed.

Line 302: The following sentence was included in the manuscript:

"The three highest RMSE"

**COMMENT:** L319: can this result also be attributed to the fact that W2 and ANN were fitted to the data and the 1D models were not?

**RESPONSE:** Thank you for your comment. Yes, the performance of W2 would be worse without calibration, and the ANN fully relied on tuning to the observational data by its definition. We mention this fact in the manuscript (Lines 169-170 and Figure 2) and discuss the representativeness of the uncalibrated 1-D models (Lines 319-321):

"This result can be attributed to the wind-forcing treatment by 1-D models. The latter do not consider the wind-sheltering effect, which was the most relevant parameter for calibration of the 2-D model, reducing the wind velocity by around 34%."

**COMMENT:** L326-7: better to refer to Fig 4d.

**RESPONSE:** Thank you. The reference was changed.

**COMMENT:** L 437: change heart to heat

**RESPONSE:** Thank you. This word was changed.

**COMMENT:** L 512: change 'average mean air temperatures' to 'mean air temperatures'

**RESPONSE:** Thank you. This sentence was changed.

**COMMENT:** Figure 4: What exactly does the trendline show? Also it is a bit difficult to compare the models with each other as they are shown in different panels. Consider plotting the individual trendlines for each model on one panel, say overlaid on the ensemble panel, if this is not too cluttered.

**RESPONSE:** Thank you. We have tried to improve the visual representation using the suggested approach. The reason for the inclusion of the trendline was to facilitate identification of the reservoirs with higher and lower RMSEs. After several editions, we decided to return to the original figure, as we felt it provided a better insight into the results.

**COMMENT:** Figure 5: caption – I don't get the first part: Evaluation of simulation bias and RMSE. 2-D baseline scenario (W2 Reservoir) simulated SWT for: a) the exclusion

of inflows and outflows (W2 Lake), ... Does it mean that you calculated bias and RMSE based on the comparison of the different model runs with W2 reservoir?

**RESPONSE:** Thank you for pointing this out. The caption was not correct. In the revised version the caption was changed to: "Evaluation of simulation bias and RMSE. 2-D baseline scenario (W2 Reservoir) simulated SWT versus: a) the exclusion of inflows and outflows (W2 Lake);"

**COMMENT:** Figure 11: suggest log scale on the x-axis.

**RESPONSE:** Thank you for the suggestion. The revised version of the manuscript includes this change.

**References**

Bedacht, E., Gulev, S. K., and Macke A.: Intercomparison of global cloud cover fields over oceans from the VOS observations and NCEP/NCAR reanalysis. *Int. J. Climatol.*, **27**, 1707–1719, doi:10.1002/joc.1490, 2007.

Calamita, E., Sebastiano Piccolroaz, S., Majone, B and Toffolon, M.: On the role of local depth and latitude on surface warming heterogeneity in the Laurentian Great Lakes, Inland Waters, 11:2, 208-222, DOI: 10.1080/20442041.2021.1873698, 2021.

Carr, M. K., Sadeghian, A., Lindenschmidt, Karl-Erich, Rinke, K. and Morales-Marin, L.: Impacts of Varying Dam Outflow Elevations on Water Temperature, Dissolved Oxygen, and Nutrient Distributions in a Large Prairie Reservoir, Environ Eng Sci, 37, 78-79, https://doi: 10.1089/ees.2019.0146, 2020.

Clark, J. V., and Walsh, J. E.: Observed and reanalysis cloud fraction. *J. Geophys. Res.*, **115**, D23121, doi:10.1029/2009JD013235., 2010.

Cole, T. M., and Wells, S. A.: CE-QUAL-W2: A Two- Dimensional, Laterally Averaged, Hydrodynamic and Water Quality Model, Version 3.6. User manual. Report of Department of Civil and Environmental Engineering, Portland State University, Portland, OR, 797, 2008.

Free, M., Sun, B., and Yoo, H. L.: Comparison between total cloud cover in four reanalysis products and cloud measured by visual observations at U.S. weather stations. Journal of Climate, 29(6), 2015–2021. https://doi.org/10.1175/JCLI-D-15-0637.1, 2016.

Jakob, C.: Cloud cover in the ECMWF reanalysis. *J. Climate*, **12**, 947–959, doi:10.1175/1520-0442(1999)012<0947:CCITER>2.0.CO;2. 1999.

Mohseni O., Erickson T.R. and Stefan H.G., 1998: A non-linear regression model for weekly stream temperatures. Water Resources Research Vol. 34, n° 10, 2685-2692, 1998.

OCDE (Eds.): Eutrophication of waters – Monitoring Assessment and control, Organization for the Economic Cooperation and Development. OECD, Paris, 154, 1982.

Stefan and Preud'homme: Relationship Between Water Temperatures and Air Temperatures for Central U.S. Streams. Project Report No. 333. St. Anthony Falls Hydraulic Laboratory. University of Minnesota, 1992.

Stengel, M., Schlundt, C., Stapelberg, S., Sus, O., Eliasson, S., Willén, U., and Meirink, J. F.: Comparing ERA-Interim clouds with satellite observations using a simplified

satellite simulator, Atmos. Chem. Phys., 18, 17601–17614, https://doi.org/10.5194/acp-18-17601-2018, 2018.

Thackston, E. L., and Parker, F. L.: Effect of Geographical Location on Cooling Pond Requirements and Performance. Water Pollution Control Research Series 16130 FDQ 03/71. Vanderbilt University, Dept. of Environmental and Water Resources Engineering. Environmental Protection Agency (EPA), Washington, D. C., 244, 1971.

Thackston, E. L., and Parker, F. L.: Effect of Geographical Location on Cooling Pond Requirements and Performance. Water Pollution Control Research Series 16130 FDQ 03/71. Vanderbilt University, Dept. of Environmental and Water Resources Engineering. Environmental Protection Agency (EPA), Washington, D. C., 244, 1971.

Toffolon, M., and Piccolroaz, S.: A hybrid model for river water temperature as a function of air temperature and discharge. Environmental Research Letters, 10(11), 114011, 2015.

Wells, S.: CE-QUAL-W2: A Two-Dimensional, Laterally Averaged, Hydrodynamic and Water Quality Model, Version 4.5. User Manual Part 2: Hydrodynamic and WaTer Quality Model Theory, 2021.

Wetzel, R.G.: Limnology, W.B. Saunders, Philadelphia, PA. 1975.

Wu, W., Liu, Y., and Betts, A. K.: Observationally based evaluation of NWP reanalyses in modeling cloud properties over the Southern Great Plains. *J. Geophys. Res.*, **117**, D12202, doi:10.1029/2011JD016971. 2012.

Wunderlich, W.: Heat and Mass Transfer between a Water Surface and the Atmosphere, Rpt. No. 14, Rpt. Publication No. 760 0-6803, Water Resources Research Laboratory, Tennessee Valley Authority, Division of Water Control Planning, Engineering Laboratory, Norris, TN, 1972.

Zhang, X., Lu, N., Jiang, H. *et al.* .: Evaluation of Reanalysis Surface Incident Solar Radiation Data in China. *Sci Rep* **10**, 3494. https://doi.org/10.1038/s41598-020-60460-1, 2020.

**Anonymous Referee 2:** In this work, Almeida et al. compared the performance of 2-D lake models with/without accounting for lateral flow, 1-D lake models (Hostetlerbased and FLake) and data-based ANN models in simulating the thermal regimes of 24 reservoirs in Portugal. They domenstrated that for reservoirs with short WRT, it is important to represent the effect of lateral flow and water level fluctuation in the lake models of GCMs and RCMs. Although the importance of lateral flow in the thermal regimes of reservoirs has been investigated by previous studies, the work of Almeida et al. is novel in three aspects: 1) the investigation of a large set of reservoirs, 2) the inclusion of ML methods, and 3) the comparison of multiple 1-D lake models. The manuscript is well written and easy to follow. I agree with the comments of the first reviewer and provide additional comments. I recommend the publication of this work after these comments are addressed.

**RESPONSE:** We would like to thank the reviewer for the time taken and for the thoughtful comments and valuable suggestions for improving our manuscript. To facilitate the work of the reviewers and editor, we will refer to the line numbers in the original manuscript when describing the revisions made.

**Major comments**

**COMMENT:** First, I possibly misunderstood but it seems that the 1-D Hostetler-based model was developed by the authors for this work. If so, I do not quite understand the rational because there are many well-tested Hostetler-based models that have already been publicly available, such as WRF-Lake. As a lake modeler myself, I worry that the development of a new model would unavoidably introduce bugs.

**RESPONSE:** Thank you for pointing out this fact. The "Hostetler-type" model was not developed specifically for this work. The model was previously applied in a PHD thesis (http://hdl.handle.net/10362/11982) and followed the straightforward Hostetler (Hostetler and Bartlein, 1990) approach. We agree that potential bugs might be occasionally introduced by re-coding even a simple model like this one but preferred to have full control over all components of at least one 1-D model. By doing so, we were able to reproduce the behavior of the Hostettler model, as implemented in various systems including WRF, and to additionally refine the model eddy diffusivity parameterization.

**COMMENT:** Second, it looks that the 1-D lake models were not calibrated in the study but the ANN model because it is based on the 2-D reservoir model was implicitly calibrated. Thus, in my view, the comparison of their performance in the current format is unfair. According to my own experience, by calibration, 1-D lake models can also mimic some effect of lateral flow and water level change. But whether this is physically sound is another story. However, my point is that the current experiment design does not convince me the superior of ANN over 1-D models in representing lake thermal dynamics for GCMs and RCMs because when we have data to train ANN we can also use the data to calibrate 1-D models.

**RESPONSE:** Thank you for this comment. We understand the reviewer's concern. In our opinion, the comparison of the models' results - 1-D models *versus* ANN - would be unfair only if the terms of the comparison were unknown. Simple models like FLake and Hostetler are coupled with numerical weather prediction models, due to their computational efficiency, but also because their parameters should not be re-evaluated when the model is applied to a specific lake. This is the principle that guided the development of both models. Also, this is the reason why the parameterization of eddy diffusion described by Hostetler and Bartlein (1990) followed the Henderson-Sellers (1985) method instead of parameterizations requiring individual model calibration (e.g., Sundaram and Rehm, 1973). It is not feasible to calibrate dozens or hundreds of lakes for numerical climate prediction. Therefore, to estimate the performance of 1-D models in the way they are applied in regional and global models we did not calibrate them during the development of this work.

At this point, in our opinion, a question needs to be answered: what is the way forward when it comes to improving on or reducing the impact of all of the abovementioned limitations, in particular the neglect of horizontal transport process?

We agree with the reviewer. Through the calibration of 1-D models it is possible to "mimic" some effect of lateral flow but in our opinion this is not the best way to address the issue. We think that, by forcing other parameters or constants, we are probably unbalancing the model's response in certain specific conditions. Moreover, as the reviewer says, we do not ensure a physically sound response. Could the solution for improving the parameterization of lakes inflows and outflows be in the consideration of a simplified hydrological model? Reducing this approach to its basics: we would compute inflows from precipitation, taking into consideration a constant runoff coefficient and a constant lake outflow. In our opinion, considering that we need to avoid the calibration of the model, this solution could also substantially increase the errors associated with surface-water temperature predictions.

This was the reason why we have included the ANN in our study. We think that progress in improving the parameterization of lakes in the climate system can be obtained by a combination of both approaches: process-based physical models and machine-learning solutions, when the limitations and advantages of each of them have been considered. It is true that the use of machine-learning approaches relies on the existence of training data that can sometimes be difficult to obtain. We think that, with the constant development of remote-sensing technologies, this limitation can be considerably diminished. It is also important to mention that, after the initial work of defining the neural network and all its components is done, the ANN needs

to be trained not calibrated, which is different. In our study we show that this approach can be a good solution for this problem.

**Specific comments**

**COMMENT:** L22-24: as indicated above, I do not think the current results can make such a statement. Further, there is another difficulty for ANN models to replace 1-D lake models in GCMs and RCMs. Compared with ANN models, 1-D lake models are much more generalized because they are physically based. For example, due to the limitation of model resolutions, usually the lake grid cells in GCMs and RCMs do not directly correspond to real lakes. We still do not know whether ANN models trained by data from real lakes can be extended to artificial lake grid cells.

**RESPONSE:** Thank you for your comment. We understand the reviewer's concerns. To clarify our point, we have included the following sentence in the revised version of the manuscript in line 24.

"Overall, results suggest that the combined use of process-based physical models and machine-learning models will considerably improve the modeling of air-lake heat and moisture fluxes."

We think that the result is quite balanced because we say that: "Our findings also highlight the efficiency of the machine-learning approach, which may overperform process-based physical models both in accuracy and in computational requirements, if applied to reservoirs with long-term observations available." We do not say that machine-learning approaches are better, only that they may perform better in certain conditions.

The heat fluxes retrieved from the output of an ANN will affect the near-surface atmospheric layer in the same way as a physically based model. The type of output of both models is precisely the same. In our opinion, the mismatch of the lake grid cells in GCMs and RCMs with the real lake dimensions is indeed a problem, but it is a problem for the physically based model - whose performance is greatly affected by the quantification of the lake maximum and mean depths. Our concern regarding the coupling of an ANN with a GCM or RCM relies more on the implementation of the training phase of the ANN. Nonetheless we believe that this constraint, with time, can be overcome.

**COMMENT:** L60: please add references to other Hostetler-based models that are involved in the ISIMIP lake sector, such as Tan et al. (2015). "Tan, Z., Zhuang, Q., & Walter Anthony, K. (2015). Modeling methane emissions from arctic lakes: Model

development and site― level study. Journal of Advances in Modeling Earth Systems, 7, 459-483."

**RESPONSE:** Thank you for pointing this out. The references were included.

**COMMENT:** L66: please cite the most recent modeling intercomparison study of Guseva et al. (2020). "Guseva, S., Bleninger, T., Jöhnk, K., Polli, B. A., Tan, Z., Thiery, W., ... &

Stepanenko, V. (2020). Multimodel simulation of vertical gas transfer in a temperate lake. Hydrology and Earth System Sciences, 24, 697-715."

**RESPONSE:** Thank you for pointing this out. The reference was included.

**COMMENT:** Table 1: Did you use the bathymetry data of the 24 reserviors to setup the models? Or did you only use mean depth, maximum depth and surface area to construct ideal bathymetry for these reserviors? Sometimes, the uncertainty in bathymetry can introduce large uncertainty in 2-D lake modeling.

**RESPONSE:** Thank you for this question. Yes, we used the bathymetry data retrieved from 1:25000 topographic charts of the future flooded watersheds area, prior to the dams' construction. We understand the reviewer's concern, as uncertainty in bathymetry can indeed affect considerably 2-D model results. The majority of the 2-D models considered here were also used for water-quality research studies which were finalized before the development of this manuscript. Therefore, they were thoroughly tested. In order to address a comment by reviewer 1, we have included the abovementioned information and a table with the grid dimensions of each reservoir.

**COMMENT:** L152-154: please rewrite this sentence. It is difficult to understand.

**RESPONSE:** Thank you for pointing this out. This sentence has been rewritten.

Line 152-153: "SWT time series were compared using statistic error measures (see Sect. 3.3 for more details), which allowed the assessment of the relation between reservoir WRT and the error that results when the advection due to inflows and outflows is neglected (as mentioned in the introduction, a common feature of contemporary GCMs and RCMs)." Was replaced with "SWT time series obtained with both scenarios, W2 hydrology and W2, were compared using statistic error measures (see Sect. 3.3 for more details), assessing the relationship between the reservoir WRT and the error resulting from the neglect of advection due to inflows and outflows (as mentioned in the introduction, a common feature of contemporary GCMs and RCMs)."

**COMMENT:** Equation 4: What is the definition of $\Phi$ ?**

**RESPONSE:** Thank you for pointing this out. This is the self-similarity function. The self-similarity of the temperature profile implies universality of the function for all lakes.

Line 261: the following sentence has been included in the manuscript:

"...and  $\Phi_T(\zeta)$  is the self-similarity function (dimensionless temperature)."

**COMMENT:** Section 3.3: I suggest adding the Kling-Gupta efficiency (KGE) as a model evaluation metric.

**RESPONSE:** Thank you this comment. We agree and understand the reviewer's suggestion. The metric was included. Hence, all tables were modified accordingly.

Line 302: the following sentence was included in the manuscript:

"...and the KGE varied from 0.61 to 0.96 ( $\bar{x}$  = 0.78; SD ± 0.09)."

Line 306: the following sentence was included in the manuscript:

"...and the KGE values varied from 0.62 to 0.76 ( $\overline{x} = 0.71$ ; SD  $\pm 0.04$ ) (Fig. 3e). The results show that a KGE value above 0.6 describes a reasonable fit between both datasets."

Line 351: the following sentence was included in the manuscript:

"Accordingly, the KGE values are above 0.96 (Table 6)."

Line 371: the following sentence was included in the manuscript:

"However, it is relevant to mention that the KGE values obtained for 1-D models indicate that, overall, they performed well (Table 6)."

**COMMENT:** L319-320: It is not true for Hostetler-based models. They can account for the wind sheltering effect, as documented in Guo et al. (2021). The difference is that the 2-D models can account for the direction effect of the wind sheltering but the Hostetler-based models cannot, which may be important for elongated reservoirs. "Guo, M., Zhuang, Q., Yao, H., Golub, M., Leung, L.. R., Pierson, D., & Tan, Z. (2021). Validation and Sensitivity Analysis of a 1― D Lake Model across Global Lakes. Journal of Geophysical Research: Atmospheres, 126, e2020JD033417."

**RESPONSE:** Thank you this comment. We understand the reviewer's concern. However, as explained above, we intentionally avoided calibration of 1-D models. Therefore, in our simulations with the 1-D models, the wind-sheltering coefficient was kept with a value of one for all simulations and wind velocity was kept unchanged.

**COMMENT: L323: delete "effect"**

**RESPONSE:** Thank you pointing this out. We agree with the reviewer and the word was deleted.

**COMMENT:** L325: This sentence is unclear to me. Do you mean the difference of RMSE between W2-reservoir and W2-lake?

**RESPONSE:** Thank you pointing this out, the reviewer is right. The sentence was modified as follows.

Line 325: "RMSE values reached..." was replaced with following sentence:

"The difference of RMSE values between W2 hydrology and W2 scenarios reached 2.7 °C, 1.2 °C and 0.9 °C, respectively (Fig. 4)."

**COMMENT:** L355: please cite Guo et al. (2021). "Guo, M., Zhuang, Q., Yao, H., Golub, M., Leung, L., R., Pierson, D., & Tan, Z. (2021). Validation and Sensitivity Analysis of a 1― D Lake Model across Global Lakes. Journal of Geophysical Research: Atmospheres, 126, e2020JD033417."

**RESPONSE:** Thank you pointing this out and for sharing this manuscript. The revised version of the manuscript includes this citation.

**COMMENT:** L369-370: I do not think it is true. As shown in Guo et al. (2021), the thermal regimes of deep lakes usually can be better simulated by Hostetler-based models than shallower lakes because deeper lakes usually have larger Wedderburn numbers. Here, the larger errors in these deeper reservoirs may be caused by other factors. For example, the default parameters, such as light attenuation coefficient, may be not suitable for these deeper reservoirs. Also, lateral flow may destablize the thermal structure of these reservoirs, making them difficult to simulate by 1-D models.

**RESPONSE:** Thank you for this comment. It is important to mention that in the manuscript we say that "HLM had a worse performance for reservoirs R3, R11, R14, R1 and for the six deepest reservoirs, R19, R20, R21, R23, R22 and R24, which indicates that the vertical heat diffusion was not optimally computed (Fig. 5b). Specifically, the explicit approximation of convective mixing in the HLM model by convective adjustment of unstable temperature profiles is apparently too rough, to simulate convective mixing in deep lakes (Bennington et al., 2014)." We are not saying that the model results are better for shallow lakes when compared with deep lakes. Reservoirs R1 and R3 are very shallow reservoirs, and reservoirs R11 and R14 are shallow when considering the depths of the other mentioned reservoirs.

The Hostetler model tended to overestimate the water-surface temperature at the same wind conditions during the entire year. This behavior is determined by the underestimation of heat diffusion to deeper layers. Perroud et al. (2009), while modeling the water temperature profiles of Lake Geneva (Maximum depth = 309 m) concluded that the Hostetler model performs well on the surface layers (0-5 m) but, due to the overestimation of the maxima squared buoyancy frequency (N2), diffusion of heat below a depth of 5m is underestimated. A similar result was described by Martinov et al., (2010). The model performed well in shallow lakes, but differences between modeled and observed water temperatures were significant in lakes with depths > 60m, due to underestimation of horizontal and vertical heat diffusion. Several authors (Subin et al., 2012; Bennington et al., 2014; Xiao et al., 2016) suggested artificially increasing heat diffusion to compensate the lack of 3-D mixing processes, when modeling the Laurentian Great Lakes.

It is also important to mention that we have replaced the eddy diffusion parameterization of the Hostetler-based model with the parameterization proposed by Sundaram and Rehm, 1973, while preserving all other default parameters (e.g., light-attenuation coefficient. This solution improved the results considerably.

Nevertheless, we agree with the reviewer: lateral flow can also contribute to the differences observed. However, to validate and quantify this fact, we would have to include this parameterization in the 1-D models and test the partial contribution of the eddy diffusion parameterization versus the lateral flow effect. A modification that will be addressed in a future study.

**COMMENT:** Section 4.2.2: please also add the computational time of 2-D models for reference.

**RESPONSE:** Thank you this comment. We agree with the reviewer and the computational time of 2-D models was added to the revised version of the manuscript.

Line 381 - Table 7 was modified in order to include the computational time of 2-D models

Line 412 - the following sentence was included:

"Table 7 also shows the significant difference in computational time between the 2-D model and all the other models."

**COMMENT:** Figure 8: The caption is confusing. I think all models use the same atmospheric forcing. For 2-D models, what this figure presents is the wind stress after accounting for the sheltering effect. Please make it clear.

**RESPONSE:** Thank you this comment. We agree with the reviewer and the caption has been modified.

Line 454 - the following caption:

"Figure 8. Mean annual wind velocity values obtained with W2 Reservoir scenarios (W2R), W2 Lake scenarios (W2L), HLM, FLake and ANN considering hourly meteorology (2005-2008). Bias between W2 Reservoir (W2R) and the other models SWT results"

Was replaced with:

"Figure 8. Mean annual wind-velocity values obtained with W2 hydrology-H (W2 hydro.-H), W2-H, (accounting for the wind-sheltering effect), HLM-H and FLake-H scenarios taking into consideration hourly meteorology (2005-2008). Bias between W2 hydrology-H and the other scenarios' mean wind-velocity values"

**COMMENT:** L483: All reservoirs tested in this study are under the Mediterranean climate. So the conclusion here is too broad. It is better to say "for the same morphometry and under the Mediterranean climate".

**RESPONSE:** Thank you for this comment. We agree with the reviewer. This change was included in the revised version of the manuscript.

**References**

Bennington, V., Notaro, M., and Holman, K. D.: Improving climate sensitivity of deep lakes within a regional climate model and its impact on simulated climate, J. Climate, 27, 2886–2911, https://doi:10.1175/JCLI-D-13-00110.1, 2014.

Hostetler, S., and Bartlein, P. J.: Simulation of lake evaporation with application to modeling lake-level variations at Harney-Malheur Lake, Oregon. Water Resour. Res., 26, 2603–2612, https://doi.org/10.1029/WR026i010p02603, 1990.

Henderson–Sellers, B. A.: New formulation of eddy diffusion thermocline models. Appl. Math. Modelling, Vol. 9, 441-446, 1985.

Martinov, A., Sushama L., and Laprise, R.: Simulation of temperate freezing lakes by one-dimensional lake models: performance assessment for interactive coupling with regional climate models, Boreal Environ. Res., 15, 143–164, 2010.

Perroud, M., Goyette, S., Martynov, A., Beniston, M., and Annevillec, O.: Simulation of multiannual thermal profiles in deep Lake Geneva: a comparison of onedimensional lake models, Limnol. Oceanogr., 54, 1574–1594, https://doi.org/10.4319/lo.2009.54.5.1574, 2009.

Subin, Z. M., Riley, W. J., and Mironov, D.: An improved lake model for climate simulations: Model structure, evaluation, and sensitivity analyses in CESM1. J. Adv. Model. Earth Sy., 4, https://doi.org/10.1029/2011MS000072, 2012.

Urolagin S., K.V. P., Reddy N.V.S.: Generalization Capability of Artificial Neural Network Incorporated with Pruning Method. In: Thilagam P.S., Pais A.R., Chandrasekaran K., Balakrishnan N. (eds) Advanced Computing, Networking and Security. ADCONS 2011. Lecture Notes in Computer Science, vol 7135. Springer, Berlin, Heidelberg. https://doi.org/10.1007/978-3-642-29280-4\_19. 2012.

Xiao, C., Lofgren, B., Wang, J., and Chu, P.: Improving the lake scheme within a coupled WRF-Lake model in the Laurentian Great Lakes. J. Adv. Model. Earth Sy. 8. https://doi.org/10.1002/2016MS000717, 2016.